

**The bacteria-protist link as a main route of dissolved organic**
**matter across contrasting productivity areas in the**
**Patagonian Shelf**
Celeste López-Abbate[1], John E. Garzón-Cardona[1,2], Ricardo Silva[3], Juan-Carlos Molinero[4],
Laura A. Ruiz-Etcheverry[5,6,7], Ana M. Martínez[8], Azul S. Gilabert[1] Rubén J. Lara[1]
[1] Instituto Argentino de Oceanografía (CONICET-UNS), Camino La Carrindanga km 7.5, 8000 Bahía Blanca,
Argentina.
[2] Universidad Nacional del Sur. Departamento de Química, 8000 Bahía Blanca, Argentina.
[3] Instituto Nacional de Investigación y Desarrollo Pesquero (INIDEP), Paseo Victoria Ocampo Nº1
(B7602HSA), Mar del Plata, Buenos Aires, Argentina.
[4] Institut de Recherche pour le Développement (IRD), UMR248 MARBEC, IRD/CNRS/IFREMER/UM, Sète
Cedex, France.
[5] Departamento de Ciencias de La Atmósfera y Los Océanos, Facultad de Ciencias Exactas y Naturales,
Universidad de Buenos Aires (DCAO, FCEN-UBA), Intendente Guiraldes 2160, Ciudad Universitaria,
Pabellón II 2do. Piso, C1428EGA Ciudad Autónoma de Buenos Aires, Argentina
[6] Centro de Investigaciones Del Mar y La Atmosfera (CIMA/CONICET-UBA), Argentina
[7] Instituto Franco-Argentino para El Estudio Del Clima y Sus Impactos (IRL-IFAECI/CNRS-CONICET-UBA),
Argentina
[8] Instituto de Química del Sur (INQUISUR-CONICET), Universidad Nacional del Sur, 8000 Bahía Blanca,
Argentina.
Correspondence to: Celeste López-Abbate (mclabbate@iado-conicet.gob.ar)





**Abstract**

While the sources of dissolved organic matter (DOM) in the open ocean are relatively well identified, its fate due to microbial activity is still evolving. Here, we explored how microbial community structure, growth, and grazing of phytoplankton and heterotrophic bacteria influence the DOM pool and the transformation of its fluorescent fraction (FDOM) during dilution experiments in the Patagonian Shelf (SW Atlantic Ocean). This area constitutes a global hotspot of carbon sequestration due to intense biological productivity which peaks at the shelf break front. The productive stations at the shelf break front featured a food web primarily based on phytoplankton and heterotrophic bacteria, while less productive mid-shelf stations showed greater dependence of protistan predators on bacterial biomass. Although phytoplankton biomass was higher than that of bacteria, protists selectively preyed on the latter, which exhibited faster growth rates, denoting high trophic specificity of grazers. Trophic efficiency and omnivory favored a bottom-heavy biomass distribution, characterized by consumer biomass dominance over producers, except in highly productive stations influenced by nutrient-rich upwelling waters, where a typical pyramid structure was observed. Our results showed that in addition to the commonly accepted factors such as phytoplankton growth stage and bacterial community composition, DOM accumulation versus consumption is also linked to bacterial grazing. Intense grazing on heterotrophic bacteria promoted DOM accumulation, likely by reducing the number of active, DOM-consuming bacteria and by providing egestion compounds to the DOM pool. Moreover, bacterial consumption of DOM appeared uncoupled from its total amount but was influenced by FDOM properties. These findings suggest that under high bacterial growth rate that follows the onset of the productive season, protistan grazers act as a link between bacterial biomass and higher trophic levels, partially diverting DOM lysate production by virus.

## 1. Introduction

Marine microbes have consensual impact on human life due to their climate-active role that stems from their ecosystem functions and broad predominance in marine biomass (Cavicchioli et al., 2019). The balance between microbial climate roles, i.e., $CO_2$ fixation, nutrient regeneration, and carbon sequestration, has consequential buffering effects on the currently unbalanced global carbon cycle (Hutchins and Fu, 2017). The prediction of microbial climate roles, however, requires a deep understanding of microbial interactions since most metabolic outcomes are shaped by both resource supply and mortality sources. Protistan grazers along with viral lysis constitute the main sources of mortality of phytoplankton and prokaryotes in the ocean (Brussaard, 2004; Calbet and Landry, 2004; Weinbauer and Peduzzi, 1995). Both mortality sources create divergent carbon routes as viral lysis directs host biomass toward the dissolved organic matter (DOM) cycle and grazing may repackage bacterial and phytoplankton biomass inaccessible to mesozooplankton and fish larvae thus linking the microbial loop with higher trophic levels (Azam et al., 1983). Protistan grazers further impact element cycle by recycling nutrients, thus prolonging bloom formation (Sherr and Sherr, 2016), and by providing DOM from excretion and egestion (Kujawinski et al., 2004). The consequences of selective grazing upon prokaryotes or phytoplankton, however, are less well understood as it may result from the interplay of various factors. For example, while size-specific grazing prompt compositional shifts in phytoplankton (e.g., Kanayama et al., 2020), the more generalist grazing on bacteria implies that community composition tends to remain more stable under grazing



pressure (Baltar et al., 2016). On the other hand, grazing on bacteria seems to remain close to bacterial
production, specially under oligotrophic conditions (Sanders et al., 1992), but phytoplankton may temporally
scape protistan grazing under favourable growth conditions thus allowing for bloom formation (Irigoien et al.,
2005). The trophic transfer efficiency varies between phytoplankton and bacteria-based food webs, with the
former typically involving fewer carbon steps before reaching microcrustaceans (Berglund et al., 2007).
DOM represents the ocean's second most significant carbon reservoir after dissolved inorganic carbon, and its
characteristics undergo alterations due to physical and biological processes on a daily basis (Spencer et al.,
2007). The optical properties of DOM offer insights into its biochemical characteristics. The chromophoric
fraction indirectly estimates phytoplankton DOM production (Romera Castillo et al., 2010), while fluorescence
serves as an indicator of its biological and photochemical reactivity (Stedmon et al., 2003). While phytoplankton
is the primary source of DOM in the ocean, other processes such as viral lysis and grazing also significantly
influence its magnitude and complexity. The impact of protistan grazing on carbon pools, is driven by the
remineralization of organic carbon and by the formation of DOM by reworking phytoplankton and bacterial
biomass (Baña et al., 2014; Lund Paulsen et al., 2019; Nagata and Kirchman, 1992). However, the impact of
selective grazing pressure on DOM-transforming prey, i.e., bacteria and phytoplankton, is less well understood.
Shelf areas represent global hotspots of carbon transformation not only due to their high productivity but also
because of the intense interaction with terrestrial habitats and the local and meso-scale mixing processes
connecting the euphotic zone with bottom sediments (Laruelle et al., 2018). The Patagonian Shelf, characterized
by highly productive frontal regions, is particularly notable for its substantial potential for carbon absorption
on a global scale (-0.02 Gt C $yr^{-1}$) (Kahl et al., 2017).The emergence of an acidification rate ranging from -
0.001 to -0.0018 per year in the water masses adjacent to the shelf is likely linked to ongoing $CO_2$ capture
processes. In particular, the northern area of the shelf exhibits the lowest pH values, attributed to the heightened
rate of remineralization occurring in the coastal region (Orselli et al., 2018). Biological processes have been
identified as primary drivers of carbon capture in the shelf area (Kahl et al., 2017; Schloss et al., 2007), with
model predictions indicating that a significant portion of autochthonous biogenic material is exported to the
open ocean in subduction zones at the confluence of the Brazil and Malvinas currents (Berden et al., 2020;
Franco et al., 2018). Despite the recognition of biological mechanisms mediated by microbial food webs as key
contributors to carbon capture in the shelf area, the underlying ecological mechanisms remain insufficiently
explored.
Given that both phytoplankton and bacteria are essential in the processing and accumulation of DOM in the
sunlit ocean and that selective grazing upon these groups impact on its subsequent directionality, we conducted
dilution experiments to measure growth and grazing of total phytoplankton and bacteria and monitored CDOM
and FDOM transformation over the course of the experiments. The aim of this study was to assess the fate of
dissolved organic matter (DOM) within the context of naturally occurring ecological interactions between
producers (phytoplankton and bacteria) and protistan grazers in two areas of the Patagonian Shelf during spring
bloom conditions. The examined areas encompassed both the mid-shelf region, characterized by low to
moderate productivity, and the shelf break, an upwelling and productive area known for recurrent spring bloom
formation. We found that regardless of productivity level, grazers preyed selectively on bacteria and that grazing



pressure on bacteria was a primary factor driving the short-term accumulation of DOM. Our results contribute
to better defining the functional roles of protistan grazers in carbon routing within the ocean.

**2.        Material and Methods**
**2.1      Sampling strategy**
The Patagonian Shelf is one of the largest continental shelf areas in the world and its more conspicuous feature
is the presence of a 2500 km-long upwelling front at the shelf break area characterized by recurrent spring
blooms and intense geochemical transformation (Romero et al., 2006). Biological carbon fixation in this frontal
area contributes substantially to the sequestration of large amounts of carbon and constitutes one of the major
$CO_2$ sinks at the global level (Kahl et al., 2017). Here, we selected two groups of stations in the mid-shelf
(stations 23, 22 and 21 from the coast to the open ocean) and the shelf break area (stations 14, 13 and 12 from
the coast to the open ocean). Mid-shelf stations were separated by ca. 30 km intercepting the 50 m isobath while
shelf break stations were separated by ca. 18 km and intercepted the 100 and 200 m isobaths. According to the
bioregionalization of the Patagonian shelf waters proposed by Delgado et al. (2023), mid-shelf stations are
located in low to moderate productivity regions (mean chlorophyll-a concentration during the spring peak
between 1.14 and 2.48 mg m$^{-3}$), while the shelf break stations are nested in the upwelling, highly productive
region (mean chlorophyll-a concentration during the spring peak of 5.8 mg m$^{-3}$). Hydrographic data
(temperature, salinity, pressure, and fluorescence) were taken with a CTD profiler SBE 9plus during the cruise
H0917 from October 9 to 12, 2017.

**2.2      Analytical determination of inorganic nutrients and DOC**
Water samples for chemical/plankton determinations and experiments, were taken from the chlorophyll-a
maximum with 6 l Niskin bottles attached to the CTD rosette, while dissolved organic carbon (DOC) samples
were taken in the surface layer (5 m). Water samples aliquots were taken for the analysis of dissolved nutrients.
The measurement of inorganic nutrients ($NO_2^-$, $NO_3^-$ and $NH_4^+$, $PO_4^{3-}$, and Si) was carried out by analyzing 50
ml aliquots of seawater preserved with $HgCl_2$ solution (Kattner and Becker, 1991) The concentration of
dissolved inorganic nitrogen (DIN) was calculated as the sum of $NO_2^-$, $NO_3^-$ and $NH_4^+$. Filtered (pre-combusted
Whatman GF/F glass fiber filters) samples for DOC were collected in pre-combusted 20 ml glass vials and
acidified to pH < 2 with $H_3PO_4$. Filtrates were analyzed using high-temperature (680°C) catalytic oxidation
with $Al_2O_3$ particles containing 0.5% platinum (Pt) in a TOC analyzer (Dohrmann DC-190, CA, USA). The
resulting $CO_2$ was then quantified using non-dispersive linearized infrared gas analysis (Skoog et al., 1997).
Potassium hydrogen phthalate solution was used as the calibration standard.

**2.3      Satellite chlorophyll-a**
Moderate Resolution Imaging Spectroradiometer (MODIS) Aqua images of chlorophyll-a concentration were
downloaded from the National Aeronautics and Space Administration (NASA) ocean color web site
(https://oceancolor.gsfc.nasa.gov/). Daily Level 3 (L3) images with a spatial resolution of 4 km were obtained
for the period spanning August 2017 to December 2017, capturing the closest pixel to each sampling point.



These images were utilized to construct time series data for each station and assess the phytoplankton's growth
phase at each location. To minimize the percentage of missing values, we computed the 5-day mean and applied
a low-pass filter to remove the variability lower than 28 days.

**2.4     Experimental set up**
Feeding experiments, based on the dilution technique (Landry and Hassett, 1982), were prepared by gently
mixing different amounts of unfiltered water and <0.2 µm water (using Whatman polycarbonate filters) in acid-
cleaned glass bottles (1 l). Four dilution treatments (D) were prepared: 10%, 40%, 70% and 100% (whole
water). An additional treatment consisting of filtered seawater (<0.7 µm, using Whatman GF/F glass fiber
filters) was set to evaluate chromophoric (CDOM) and fluorescent dissolved organic matter (FDOM)
modifications in the absence of protists and grazers. Seawater from each site was obtained from the chlorophyll-
a maximum (20-30 m depth), and pre-filtered by a 200 µm mesh net to eliminate large, metazoan grazers.
Experimental bottles (3 replicates) were daily (24 h) deployed at a deck-incubator (200 l) equipped with
continuous in situ water flow and covered with a double knitted mesh fabric (~215 g m$^2$) to attenuate the UV
radiation. Dissolved inorganic nutrients (N, P and Si) were added to the incubation bottles by assuming a
maximum chlorophyll-a concentration in the mid-shelf area of 2 µm l$^{-1}$, and in the shelf-break area of 10 µm l$^{-1}$
$^1$ (Delgado et al., 2023). The amount of nutrients added followed Calbet and Saiz (2018) to ensure
phytoplankton growth at non-limiting conditions. A series of dilution bottles without nutrient addition was set
as control treatment.

**2.5     Phytoplankton and bacterial growth and grazing by phagotrophic protists**
Rate estimates of phytoplankton growth (µ) and mortality due to protist phagotrophy (m) were obtained using
the equations of Landry and Hassett (1982). While initially intended for measuring phytoplankton growth and
mortality, we adapted this method to assess bacterial growth rate and bacterivory. This approach has been
demonstrated to be effective and reliable for use with natural bacterial communities in non-oligotrophic regions
(Tremaine and Mills, 1987). The method is based on measuring the initial and final concentration of
chlorophyll-a (as a proxy of phytoplankton biomass) and bacterial abundance in triplicate dilution series after
an incubation period of 24 h. It assumes that protistan grazing rate is a linear function of prey concentration
(Holling type I functional response), and can be calculated as follows:

$\mu_0 = 1/t \ln Pt/P0 = \mu - mD$

where $\mu_0$ is the apparent growth rate, P0 and Pt are the phytoplankton concentration at the initial (0) and final
(t) conditions, respectively, D is the dilution series. We tested model fit in every experiment.

**2.6     Plankton abundance and biomass**
Subsamples from the initial and final treatments were collected for chlorophyll-a and bacteria abundance
analysis. To determine chlorophyll-a, triplicate samples (300 ml) were filtered through GF/F filters and stored





at -20°C. Pigments were extracted with 90% acetone for 24 h in the dark at -20° C and then determined
spectrophotometrically according to Jeffrey and Humphrey (1975). Triplicate samples of picoplankton (3 ml)
and duplicate samples of nanoplankton (100 ml) were fixed with 0.53 ml of glutaraldehyde (f.c. 2 %) and
subsequently processed following the methods described by Porter and Feig (1980). Heterotrophic bacteria
were quantified by staining 1 ml seawater sample with 4,6-diamidino-2-phenylindole (DAPI) to a final
concentration of 3 µg ml$^{-1}$ and collected on black polycarbonate filters (25 mm diameter, 0.2 µm pore size).
The enumeration was done with a microscope Nikon Eclipse 80i equipped with a fluorescence lamp at 100X
magnification. Heterotrophic bacteria were identified using a UV excitation filter (330-385 nm). Twenty-five
images were taken at random points from each polycarbonate filter using a Nikon DXM1200F digital camera
and subsequently, every cell in the image was enumerated and sized using the software ImageJ. Bacterial cell
volumes were calculated assigning simple geometric shapes to species (coccos, bacillus), and converted into
carbon content (µg C l$^{-1}$) by the allometric model according to Simon and Azam (1989).
Protist plankton identification and quantification were conducted using light and epi-fluorescence microscopy
at the initial treatment stage. The identification of photosynthetic (PNP) and heterotrophic nanoplankton (HNP)
was done by a combination of light and epi-fluorescent microscopy. Note that the size categories PNP and HNP
include members from nanoplankton (5-20 µm) and ultraplankton (>5 µm). Prior to cell enumeration, preserved
samples (5 ml) were stained with DAPI (f.c. 5 µg ml-1) and proflavin (f.c. 5 µg ml-1) and collected on black
polycarbonate filters (25 mm diameter, 0.2 µm pore size). Most PNP and HNP were identified using a blue
excitation filter (450-490 nm) while Cryptophytes were identified using a green excitation filter (480-550 nm).
Cell enumeration was done by settling the preserved sample (1-2 ml) in Utermöhl chambers during 24 h. The
entire chamber was analyzed under a Wild M20 inverted light microscope. Similarly, the enumeration of
phytoplankton and phagotrophic protists in the size fraction 20-200 µm was done by settling a variable volume
(10–50 ml, depending on sediment and plankton concentration) of preserved seawater sample (Lugol's iodine)
in Utermöhl chambers during 24 h. It is worth mentioning that samples were pre-filtered through a 200 µm
mesh to exclude larger consumers from our experiments. This procedure may have also removed colony-
forming protists. Biomass estimation involved assigning simple geometric shapes to species to quantify cell
volume, which was subsequently converted into carbon content (µg C l$^{-1}$) according to Hillebrand et al. (1999).
Protistan taxa abundance was visualized by a heatmap (employing the R package *heatmaply*), and taxa was
segmented into functional groups to facilitate visualization. A side dendrogram was included to group similar
sampling stations by ordering rows (stations) so that the sum of distances between each one will be minimized.
Data for ranking rows was normalized to range from 0 to 1. To assess the dominant taxa contributing to station
ordination, a biplot based on non-metric Multi-Dimensional Scaling (MDS) was done to evaluate the correlation
of taxa on the station ordination using the R package *vegan*.

**2.7    Production of CDOM and FDOM transformation**
Given that protistan grazing impacts on carbon pools, either by remineralizing organic carbon or by contributing
with the formation of DOM by reworking phytoplankton and bacterial biomass (Baña et al., 2014; Lund Paulsen
et al., 2019), and that these processes add to the isolated effects of phytoplankton growth and bacterial



degradation of DOM in natural communities, we estimated the net DOM production and evaluated the fate of
biodegradable and biorefractory compounds during the incubations. For this purpose, we measured CDOM and
FDOM at the beginning and end of the experiments in the presence (undiluted treatment, prefiltered by 200
µm) and absence of protists (undiluted treatment, prefiltered by 0.7 µm). This procedure captured
transformation processes within closed systems at the daily basis. Many biotic and abiotic transformation
processes occur at the daily timescale. For instance, photodegradation of refractory products occurs within hours
(Timko et al., 2015), while experimental observations revealed that significant shifts driven by biological
processes were identifiable after 24 h (Lønborg et al., 2010, 2015; Urban-Rich et al., 2004). Furthermore,
experimental results revealed that major microbially driven DOM transformation occur within the first 24 h
upon their release by phytoplankton (Gruber et al., 2006; Hach et al., 2020). This emphasizes that detectable
DOM transformation processes occurring at short-term periods, can provide clues to assess transient DOM
trends within specific succession phases of microbial communities. It is worth mentioning that the treatment
filtered by 0.7 µm, excluded part of the bacterioplankton community, notably the particle-attached bacteria, and
thus may not accurately reproduce the response of natural communities.
The optical properties of FDOM were evaluated from emission-excitation matrices (EEM) obtained with a
Shimadzu RF-5301 scanning spectrofluorometer with a 150W xenon lamp and a 1 cm quartz cell. Milli-Q water
was used as reference and the intensity of the Raman peak was regularly checked. The emission wavelength
ranged between 250 nm and 600 nm while the excitation wavelength ranged between 220nm and 370nm. An
estimation of dissolved humic-like and protein-like substances was carried out at the wavelengths proposed by
Coble (1996). Humic-like fluorophores: $FDOM_C$, containing mostly highly unsaturated components, at Ex/Em:
350/440 nm; $FDOM_A$, with moderate degree of unsaturation, at Ex/Em: 250/425 nm and $FDOM_M$, with low
degree of unsaturation, at Ex/Em:310/380 nm. Protein-like fluorophores: $FDOM_T$, with fresh components, at
Ex/Em: 270/330 nm and $FDOM_B$, corresponding to DOM transformed by biological or physicochemical
factors, at Ex/Em: 260/300 nm. Fluorescence intensity of fluorophores was expressed in Arbitrary Units (AU).
Fluorophores were identified using the PARAFAC multivariate algorithm (Stedmon and Bro, 2008) and
different biogeochemical indicators such as humification index (HIX), fluorescence index (FI), and freshness
index (BIX), were calculated (Coble, 1996). The HIX serves as a tool for assessing the diagenetic condition of
DOM, as it increases with aromaticity (Bai et al., 2015), while the FI distinguishes between DOM of different
origins, i.e., terrestrial vs. microbial (McKnight et al., 2001). BIX aims to estimate the relative contribution of
DOM produced in situ by microbes (Huguet et al., 2009).
CDOM was measured as the absorbance spectra between 240 to 800 nm measured in a Perkin Elmer Lambda
35 spectrophotometer. The absorbance at 254 nm (a254) was used as a proxy of total DOM in the UV spectrum
(Brandstetter et al., 1996). Net DOM production was calculated as 1/t ln (a254)t/(a254)0, where (a254)0 and
(a254)t are the absorbance at 254 nm of CDOM at the initial (0) and final (t) conditions, respectively. Lee et al.
(2018) identified parameters with more than a 50% absolute percent difference between the control and treated
samples as reliable indicators to distinguish between DOM transformation caused by biodegradation, UV
irradiance, and adsorption. Here we used the tendency during incubation of BIX, HIX, FI, and the ratio between
FDOM components M and A ($FDOM_M/FDOM_A$) as reliable parameters for the discrimination of





biodegradation versus UV photodegradation or adsorption. Pairwise relationship between net production of
DOM in the presence and absence of protists with variables of interest, was evaluated by simple regression
models. The same procedure was used with other variables to test for pairwise relationships of ecological
significance.

**3.      Results**
**3.1      Phytoplankton phenological stages at the sampling area**
Mean surface chlorophyll-a, derived from satellite observations during the sampling period (October 9-12,
2017, Fig. 1a), revealed a band of high phytoplankton concentration at the shelf break, centred at the 114 m
isobath in the latitudinal band at 40°S. While the spring bloom typically begins during September in the
latitudinal range of our sampling area (Delgado et al., 2023), phytoplankton at the time of sampling, as estimated
from satellite chlorophyll-a, were at different phenological stages in each station (Fig. 1b). At the mid-shelf,
station 21 showed the highest concentration of satellite chlorophyll-a and the phytoplankton community was at
the pulse initiation. Stations 22 and 23 showed lower chlorophyll-a and were sampled at bloom stationary phase.
At the shelf break, satellite-derived chlorophyll-a levels were elevated in stations 13 and 14, indicating
proximity to the bloom peak, whereas station 12 exhibited low chlorophyll-a concentrations, corresponding to
the bloom termination phase.

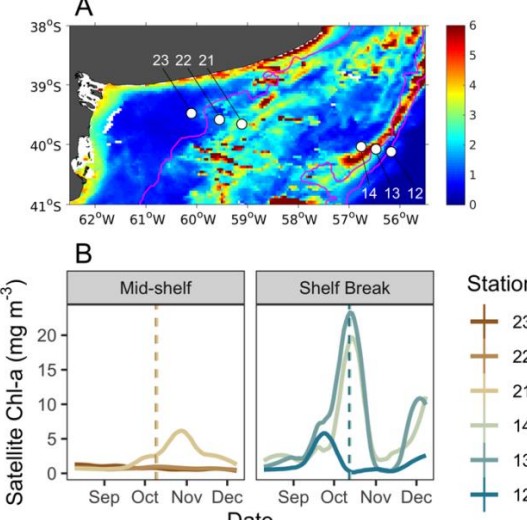


**Figure 1.A. Map of the study site showing the location of sampling stations and the mean surface**
**distribution of satellite chlorophyll-a (MODIS-AQUA) during the sampling period (October 09-12). B.**
**Temporal evolution of surface satellite chlorophyll-a concentration at grid points closest to sampling**
**stations. Solid lines denote satellite chlorophyll-a concentration at each station, while dotted lines**





**represent the moment of in situ sampling. Lines are color-coded according to the points representing**
**each station.**
**3.2      Hydrography, nutrients, and DOM properties**
Thermohaline signature was in the range of the Subantarctic Shelf waters (33.5 < S < 34) from station 21 to
station 14 in agreement with Berden et al. (2020) and Ferronato et al. (2023). At the mid-shelf area, station 23
and 22 showed relative higher salinity water (S > 33.7), linked to the coastal maximum salinity waters originated
at San Matias Gulf (Lucas et al., 2005). These stations also showed weak stratification, while the rest of the
stations showed a sharper thermocline. The mixed layer depth (MLD) ranged between 10 (stations 12 and 23)
and 31 m (stations 13 and 22). Stations 14 and 21 showed intermediate MLD of 30 and 28 m respectively. All
samples taken at the chlorophyll-a maximum were positioned within the mixed layer.
The concentration of dissolved nutrients (DIN, $PO_4^{3-}$ and Si) was highest at station 14 while the lower total
nutrient concentration was recorded at station 23 (Fig. 2a). The primary nitrogen source was $NO_3^-$, except for
station 22 where $NH_4^+$ predominated. The only notable distinction between station groups was the concentration
of $NO_3^-$, which averaged 1.2 µM in the mid-shelf stations and 7.3 µM in the shelf-break stations. According to
Redfield ratios (Redfield et al., 1963), a strong nitrogen depletion in relation to $PO_4^{3-}$ and Si occurred in station
23. While the N:P ratio was closer to 16:1 in the rest of the stations, a general excess of $PO_4^{3-}$ in relation to DIN
was registered. On the contrary, all station except station 23, showed a Si depletion in relation to DIN. The
concentration of DOC was homogeneous in the mid-shelf stations (mean of 79 µM), while in the shelf break it
varied from 96 µM in station 14 to 52 µM in station 13 (Fig. 2b). The highest fluorescence intensity of protein-
like compounds was found in station 23, while the highest intensity of humic-like fluorophores was observed
in station 21 (Fig. 3). The a254 was higher in the mid-shelf stations (mean of 2.3 $m^{-1}$) compared to the shelf
break stations (mean of 1.3 $m^{-1}$).

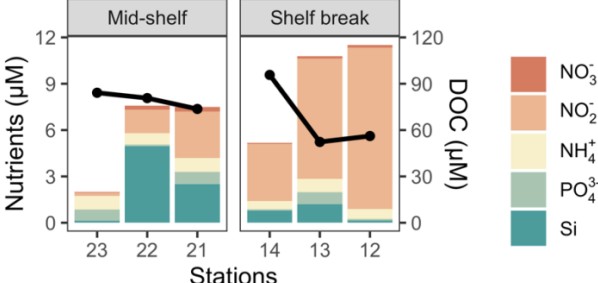

**Figure 2. Cumulative nutrient concentrations at the deep chlorophyll-a maximum (bars) and the**
**concentration of DOC in surface waters (solid black line) across stations.**



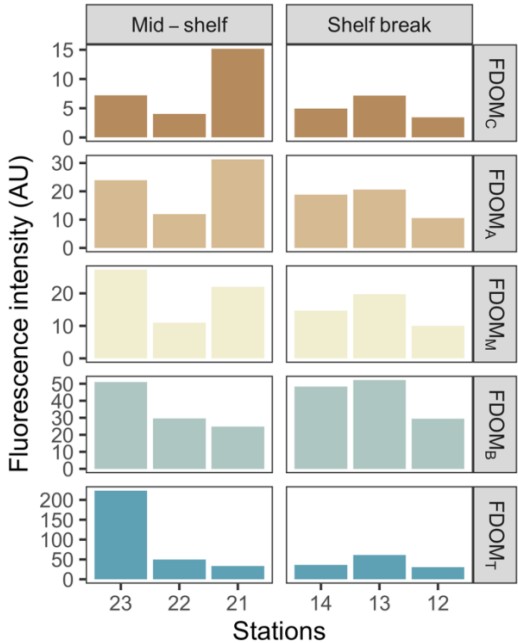


**Figure 3. Fluorescence intensity of main identified FDOM components at the deep chlorophyll-a maximum across stations. FDOM components are shown in a decreasing order of humification from top to bottom plots.**


### 3.3 Plankton community structure

The type of food web structure based on the ranking of the carbon biomass of phagotrophs, phytoplankton and heterotrophic bacteria, and the spatial distribution of these groups' biomass are shown in Fig. 4a and b, respectively. Biomass of heterotrophic bacteria ranged between 2.6 (station 22) and 15 µg C l$^{-1}$ (station 14). The abundance of this group was positively associated with chlorophyll-a concentration ($R^2$=0.7, p=0.04) and the abundance and biomass of most phytoplankton groups (p<0.05), except coccolithophores. The highest bacterial abundance and biomass was registered under chlorophyll-a pulse initiation (stations 21, 13 and 14). Among phagotrophic protists, the most significant group in terms of biomass were dinoflagellates ranging from 0 (station 14) to 134 µg C l$^{-1}$ (station 22, mostly due to the presence of *Noctiluca scintillans*). Ciliates ranged from 0 (station 14) to 20 µg C l$^{-1}$ (station 21), while HNP showed the highest biomass in stations 13 and 14 (5 and 6 µg C l$^{-1}$, respectively), and the lowest value was registered in station 12 (0.6 µg C l$^{-1}$). Ultrazooplankton (choanoflagellates and other unidentified flagellates) was the dominant fraction among HNP except in station 13, were micro-sized ciliates and nano-sized flagellates (*Telonema* sp. and unidentified dinoflagellates) dominated biomass.

324





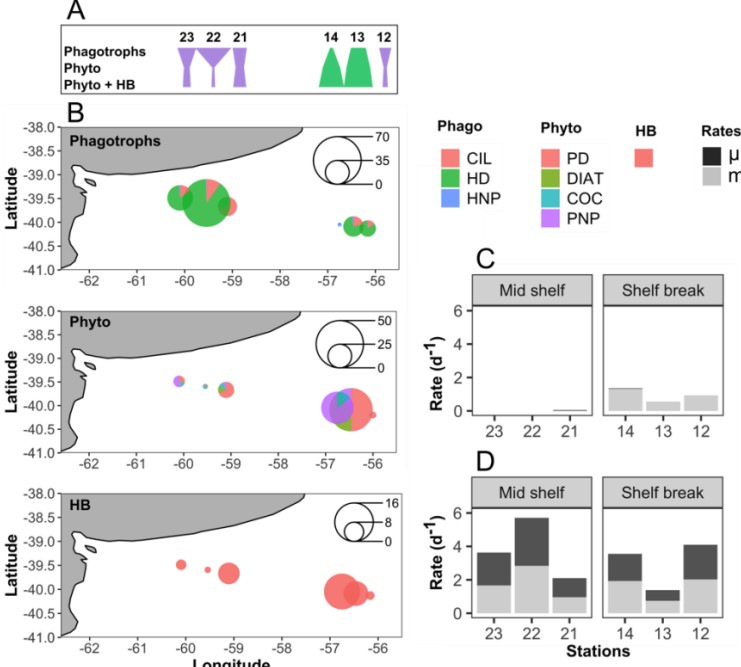

**Figure 4.A. Type of food web structure based on the ranking of carbon biomass of phagotrophs, phytoplankton (phyto), and heterotrophic bacteria (HB). B. Spatial distribution of the cumulative biomass (µg C l⁻¹) of phagotrophs (upper plot), phytoplankton (mid-plot) and HB (lower plot). Scale circles are shown within each plot. C. Growth (µ) and grazing (m) rates of phytoplankton. D. Growth (µ) and grazing (m) rates of HB.**

Whitin phytoplankton, four main groups were identified: photosynthetic nanoplankton (PNP, including photosynthetic nanoflagellates, nano-sized diatoms and dinoflagellates), micro-sized photosynthetic dinoflagellates (PD), micro-sized diatoms, and coccolithophores. PNP generally dominated the biomass of photosynthetic taxa. The highest concentration and biomass of all groups, except for coccolithophores, was registered in station 13 (PNF: 32 µg C l⁻¹, PD: 8 µg C l⁻¹, diatoms: 6 µg C l⁻¹). High biomass of PNP and dinoflagellates was also registered in station 14 (34 µg C l⁻¹) and station 21 (8.5 µg C l⁻¹), respectively. The highest biomass of coccolithophores was registered in station 22 (2 µg C l⁻¹).

Stations 12 and 13 showed conspicuous differences on microplankton community structure compared to the rest of the stations (Fig. 5). According to the ordination fit between vectors (i.e., taxa) and stations, the phagotrophic species that mainly contributed to separate these two stations from the others were the dinoflagellates *Gymnodinum* spp., and *Protoperidinium pellucidum*, while *Pyramimonas* sp. and *Dinophysis acuminata*, were the distinctive photosynthetic species in these stations (MDS, p<0.05). Stations 22 and 23 were also closely associated regarding the heterotrophic community and the species that contributed most to this association was *Strombidinopsis* sp.



346

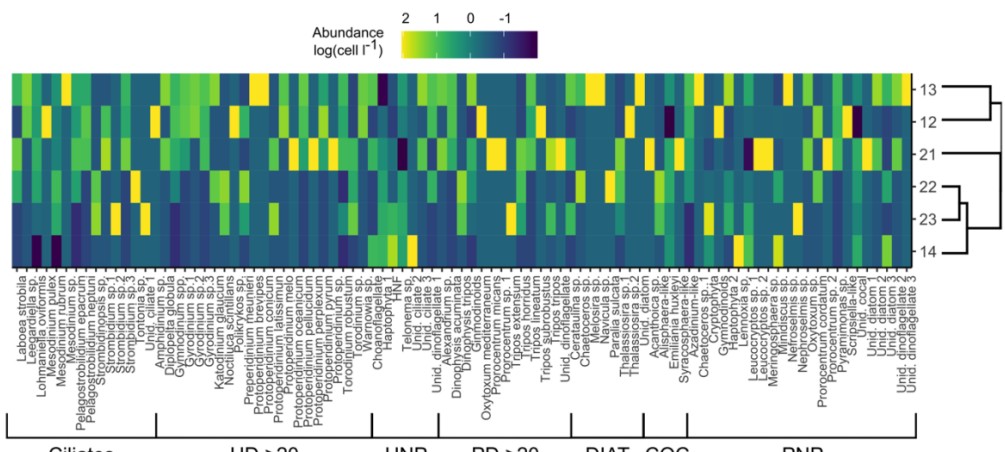

347

**Figure 5. Color-coded, log-transformed cell abundance (cell l⁻¹ × 10³) of plankton taxa (columns) at the sampling stations (rows). Functional groups delimitation is indicated in the bottom. Side dendrogram shows the optimal ordering of rows (stations) so that the sum of distances between each one is minimized. HD>20: Heterotrophic dinoflagellates >20 µm, HNP: Heterotrophic nanoplankton, PD>20: Photosynthetic dinoflagellates >20 µm, Diat: Diatoms, Coc: Coccolithophores, PNP: Photosynthetic nanoplankton.**

### 3.4 Growth and grazing rates

Water temperature of the incubator container was hourly monitored and ranged between 12.2 and 13.8°C in the mid-shelf stations and between 8 and 11.1°C in the shelf break stations. Bacteria showed active growth during all experiments while phytoplankton only revealed significant growth rates in conditions of pulse initiation (stations 21 and 14). Some degree of nutrient limitation was detected in the mid-shelf stations as the growth rate of phytoplankton at the control treatments was lower than in the nutrient amended treatment, however, differences were not statistically significant. No apparent differences were found in stations 12,13 and 14. Significant grazing effect on bacteria was found in all experiments (Fig. 4c), while grazing on phytoplankton was only significant in the shelf break stations (Fig. 4d). In the mid-shelf stations, daily bacterial productivity consumed by HNP averaged 72%, while in the shelf break stations, it reached 83%. Mean daily primary productivity consumed by phagotrophic protists was zero in the mid-shelf stations and 155% in the shelf break. Linear responses were found in all experiments, indicating that no cascading effects, saturating feeding and or starvation occurred within incubation bottles. The abundance of heterotrophic bacteria was negatively correlated with the grazing of HNP ($R^2$=0.8, p=0.016) and growth ($R^2$=0.9, p=0.005).

369



**3.5    Short-term DOM transformations**

DOM accumulation was observed in stations 22, 14, 13, and 12 during the incubation period in the experimental setting with protists, while stations 23 and 21 exhibited DOM consumption (Fig. 6a). Conversely, in the experimental setting without protists, DOM accumulated in stations 22, 21, 14, and 12, while stations 23 and 13 showed DOM consumption (Fig. 7a). Regardless of net DOM production, biodegradation of organic matter occurred in most experiments as denoted by the decrease of the BIX and the increase on the HIX (Fig. 6b, 7b). The prevalence of biodegradation over UV photodegradation or adsorption was further supported by most source discrimination indices (FI and M/A, data not shown). The decrease on BIX was coherent with the net zero to low phytoplankton growth during our experiments. An exception to this general pattern occurred in station 13, in which HIX increased and BIX decreased during the incubation. Station 13 was characterized by the highest abundance of micro-sized phytoplankton (diatoms and photosynthetic dinoflagellates) and registered the lowest concentration of DOC (52 µM) at the moment of sampling. While HIX and BIX indices suggested that DOM modifications are not driven by biodegradation, M/A decreased and FI increased during the experiments, thus giving inconsistent results.

Among treatments, a shift from DOM accumulation to consumption was registered in stations 21 and 13. In the presence of protists, the net DOM production was negatively associated with the fluorescence intensity of peaks related to humic-like compounds (FDOM$_C$: $R^2$=0.6, p=0.07, FDOM$_A$, $R^2$=0.8, p=0.01, FDOM$_M$: $R^2$=0.7, p=0.04), and positively associated with the grazing on HB ($R^2$=0.7, p=0.00). In the absence of protists, the net DOM production was negatively associated with the ratio between bacteria and phytoplankton biomass ($R^2$=0.9, p=0.01) (Fig. 8).

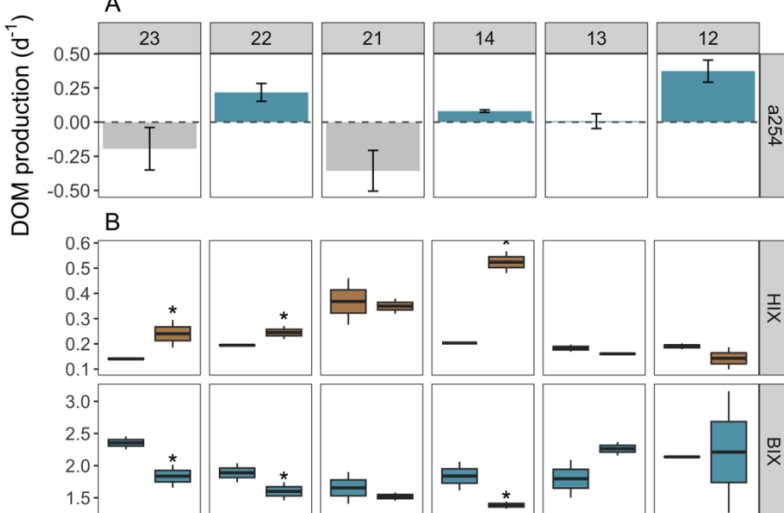





**Figure 6. DOM transformations in the experimental setting with protists (prefiltered by 200 μm) across stations. A. Net DOM production as depicted by the absorbance intensity at 254 nm, a proxy of total DOM concentration. Dashed line indicates the limit between net DOM consumption (negative values) and net DOM accumulation (positive values). B. Shifts in the humification index (HIX) and the biological activity index (BIX) during the 24-h incubation. Asterisks indicate significant differences identified by linear regression analysis between initial and final treatments (p<0.05).**

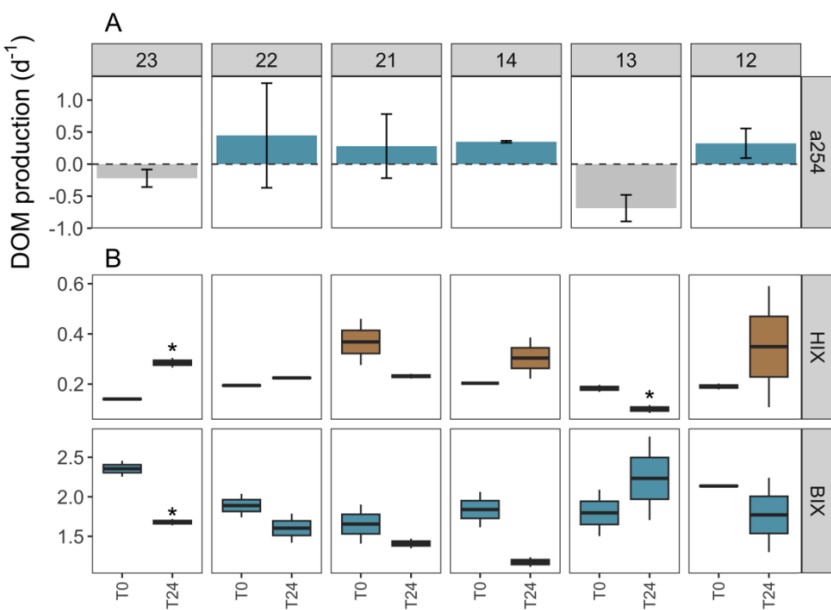

**Figure 7. DOM transformations in the experimental setting without protists (prefiltered by 0.7 μm) across stations. A. Net DOM production as depicted by the absorbance intensity at 254 nm, a proxy of total DOM concentration. Dashed line indicates the limit between net DOM consumption (negative values) and net DOM accumulation (positive values). B. Shift in the humification index (HIX) and the biological activity index (BIX) during the 24-h incubation. Asterisks indicate significant differences identified by linear regression analysis between initial and final treatments (p<0.05).**

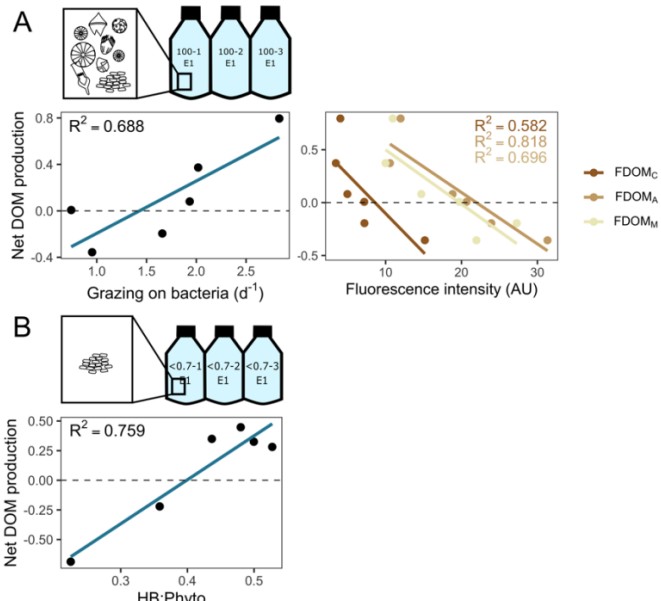

**Figure 8. Main predictors of the net dissolved organic matter (DOM) production during incubations. A. Experimental setting with protists. Linear regression plots depict the relationship between net DOM production and grazing on bacteria, and humic-like substances (as depicted by fluorophores FDOM$_C$, FDOM$_A$, and FDOM$_M$). B. Experimental setting without protists. Linear regression plot depicts the relationship between net DOM production and the ratio between heterotrophic bacteria (HB) and phytoplankton (Phyto) biomass. Dashed lines indicate the limit between net DOM consumption (negative values) and net DOM accumulation (positive values).**

## 4    Discussion

### 4.1    Microbial food web structure

The composition and distribution of the plankton community varied notably between blooming and non-blooming stations across our study area. Station 14 separated from this ordination due to its unique plankton structure resembling non-blooming areas, despite its location within a blooming region. This anomaly may be attributed to the dominance of a single nanoplanktonic diatom species, which accounted for a significant portion of the photosynthetic biomass, alongside a predominantly ultra and nano-sized protistan grazer community. Moreover, station 14 exhibited the highest bacterial biomass, suggesting a prevalence of small planktonic organisms favored by resource stoichiometry characterized by low $PO_4^{3-}$ and Si concentrations but high surface DOC. Furthermore, the location of station 14, coastward of the 100 m isobath and outside the Malvinas Current jets, indicates that nutrient-rich upwelling events at this site may be intermittent (Franco et al., 2008).

The analysis of biomass distribution among photosynthetic plankton revealed the predominance of ultra- and nanophytoplankton, with solitary diatoms, cryptophytes, haptophytes, and *Azadinium*-like dinoflagellates being



the most notable representatives within these size classes. Similar findings were reported in previous studies by
Negri et al. (2013, 2016) and Silva et al. (2009), indicating the widespread dominance of ultraphytoplankton,
primarily *Synechococcus*, in the region. Although cyanobacteria were not quantified in our study, they are
assumed to play a crucial role in carbon fixation and immobilization, particularly during warm months. In the
shelf break area, nano-sized phytoplankton, particularly diatoms, were dominant, as reported by Carreto et al.
(2016). Micro-sized phytoplankton biomass was mostly comprised of photosynthetic dinoflagellates, followed
closely by diatoms, consistent with findings by Ferronato et al. (2021, 2023), in the mid-shelf and shelf break
areas during spring conditions.
Dinoflagellates also dominated the biomass of phagotrophic protists, with *Noctiluca scintillans* and nano-sized
dinoflagellates being the primary contributors, as observed by (Carreto et al., 2016) in the shelf break area.
Ciliates, particularly nano-sized taxa, were the second most important contributors to heterotrophic biomass,
followed by micro-sized ciliates such as *Laboea strobila*. No clear distribution patterns in protistan grazers were
observed, a phenomenon well-documented in marine habitats associated with factors such as compositional
turnover, dispersal ability, differential environmental response, and interspecific interactions, coupled with
local bloom timing (Grattepanche et al., 2016; Péquin et al., 2022; Snyder et al., 2021; Zhao et al., 2022).
Mesoscale and submesoscale structures in the Patagonian Shelf create spatial heterogeneity, impacting the
distribution of dissolved resources (Garzón-Cardona et al., 2021) and chlorophyll-a (Becker et al., 2023;
Saraceno et al., 2024). The stirring produced by these processes, while understudied at the local scale, is
fundamental in driving the three-dimensional distribution of plankton (Lehahn et al., 2018). Furthermore, the
mosaic distribution of nutrients in turbulent areas may alter bacterial composition, with different nutrient
affinities potentially reshaping the entire microbial interactome (Delgadillo-Nuño et al., 2024).
The abundance and biomass of heterotrophic bacteria fell within the reported minimum values for spring and
summer in the study area, particularly resembling values typically observed during summer in the coastal zone
(<50 m depth) (Garzón-Cardona et al., 2021; Hozbor et al., 2013; Negri et al., 2016). While prior studies in this
sector did not establish a clear link between heterotrophic bacterial abundance and chlorophyll-a, it is commonly
noted that bacteria thrive in frontal regions with high phytoplankton concentrations and secondary production.
In addition, previous research has indicated a positive correlation between bacterial abundance and dissolved
organic carbon (Garzón-Cardona et al., 2021). In our study, we did not observe significant relationships between
bacteria and DOM proxies (i.e., a254), while we did find a significant positive relationship with chlorophyll-a.
In turn, bacterial abundance exhibited a negative response to high HNP grazing pressure. This suggests that
during periods of high productivity in spring, bacterial distribution is influenced not only by the supply of fresh
DOM from phytoplankton but also by grazing-induced mortality. The intermittent coupling between bacteria
and phytoplankton and the lack of reported relationships between bacteria and inorganic nutrients also suggest
that bacteria are weakly resource-controlled in this area. Instead, considering that the control of grazing on
bacterial biomass is more pronounced under conditions of low bottom-up regulation (Morán et al., 2017), our
results suggest that under the examined conditions bacteria are top-down controlled, with maximum attainable
biomass limited by grazing pressure as well as other unidentified sources of mortality.



### 4.2 Microbial trophic pathways

Our findings are in line with previous observations in the region, emphasizing the tight trophic coupling and minimal sinking of unused prey biomass (Negri et al., 2013). This is attributed to the spatial uniformity of the trophic network composition, characterized by the dominance of small phytoplankton (<5 µm) and micro- and mesozooplankton groups, which exhibit comparable biomass magnitudes. Indeed, group-specific biomass distribution of <200 µm plankton in this study, revealed that except for station 13 and 14 that were at the bloom peak, mid-shelf stations and station 12 (post-bloom) showed a top-heavy biomass distribution or inverted pyramid. Drivers of top-heaviness are linked to high trophic transfer efficiency, faster turnover of prey than consumers, or omnivory that bypasses inefficient trophic levels (McCauley et al., 2018). Since the turnover of phytoplankton and bacteria is similar to that of protistan grazers, the most likely reason for the inverted pyramid structure is a high trophic efficiency and the presence of omnivore consumers. Protists are known to be highly efficient feeders (Weisse et al., 2016), particularly under high prey abundance, and trophic transfer is significantly increased by the presence of mixotrophs (Ward and Follows, 2016). Although mixotrophy was not directly assessed in this study, the predominance of nanoflagellates among protistan grazers implies a potential significant role of mixotrophy in shaping the trophic structure of the microbial community (Edwards, 2019). In contrast, the classical bottom-heavy pyramid biomass structure of plankton was registered under bloom peak at shelf break stations 13 and 14. Despite intense predation on phytoplankton in the shelf break stations, the biomass accumulation resulting from a higher carrying capacity, driven by local upwelling, was sufficient to maintain the typical biomass pyramid structure, with photosynthetic taxa dominating plankton biomass. This type of food web structure attains the highest carbon biomass regardless of community composition (Kang et al., 2023), and suggests that under productive conditions within the upwelling front, a fraction of primary production escapes predation by protists and is either exploited by microcrustaceans or advected by mixing or sinking processes.

Our results denoted active growth of heterotrophic bacteria in all stations, while phytoplankton evidenced low net growth or biomass yield. Low growth rate of phytoplankton was only detected in station 21 at the bloom initiation and in station 14 at the last stage of the ascending bloom ramp. Despite possessing half the biomass of phytoplankton, grazers selectively preyed upon heterotrophic bacteria and accounted for 72% of bacterial production in the mid-shelf stations while phytoplankton production was not significantly affected by grazing. This selective grazing on bacteria can likely be attributed to their higher growth rate compared to that of phytoplankton, as protistan grazing is known to be activated in response to changes in prey availability (Banse, 1982; Chen et al., 2009). In fact, grazing by HNP showed a negative relationship with bacterial biomass, denoting that prey growth is a better activator than their biomass. Consequently, bacteria exhibited growth advantages over phytoplankton across the examined environmental gradient but were concurrently more vulnerable to grazing pressure. This compensatory grazing on fast-growing bacteria has been previously observed in productive environments such as the California Current (Goericke, 2011; Landry et al., 2023; Taylor and Landry, 2018a) as well as in the oligotrophic Warm Taiwan Current (Chiang et al., 2014). The suggested mechanism underlying this trophic interaction posits that increased phytoplankton-DOM production fosters the growth of resource-efficient bacteria as suggested by the tight coupling between bacteria and



phytoplankton biomass. However, the heightened growth, coupled with a diminished allocation of energy to
defensive skills, renders these bacteria susceptible to selective grazing (Taylor and Landry, 2018a).
The situation in the more productive shelf break area, shifted toward a coupled predation upon both prokaryotes
and phytoplankton (Fig. 9a). Grazing accounted for 83% of bacterial production and 154% of phytoplankton
production. Despite limited growth, phytoplankton faced substantial grazing pressure, possibly facilitated by
shared predators with bacteria. This trophic interaction aligns with the enhanced microbial loop hypothesis
(Taylor and Landry, 2018b), suggesting that small phytoplankton is increasingly grazed as a byproduct of
grazers actively preying on bacteria under conditions of rising productivity. Under this scenario, grazing on
picophytoplankton is density-independent and occurs due to the presence of shared common grazers with
bacteria. Across stations, density-dependent control mechanisms further regulated the standing stock of
bacteria, with mechanisms reducing population density likely conferring benefits by preventing rapid resource
depletion. Overall, bacteria appeared to be positively regulated by commensalism with phytoplankton and
negatively by grazing, constituting a primary carbon source for protistan grazers regardless of the productivity
level.
**4.3      Short-term DOM pathways**
A general trend denoting weak DOM limitation of bacterial growth was found, as even in the experiments
without DOM-producing phytoplankton, organic matter accumulated in most experiments. During the
experiments, high bacterial growth, and net zero or low phytoplankton growth coincided with active
biodegradation of organic matter, as indicated by the decline in the biological index (BIX) and the rise in the
humification index (HIX). The general decreasing trend of BIX during the incubation and its similar behaviour
in the absence of protist confirms a bacterial effect and suggest that bacteria are rapidly consuming recently
produced phytoplankton exudates. Similar observations have shown bacterial production of humic-like
substances from protistan plankton precursors under experimental conditions devoid of terrestrial influence
(Gruber et al., 2006; Kinsey et al., 2018; Lechtenfeld et al., 2015; Osburn et al., 2019; Romera-Castillo et al.,
2011), underscoring the significance of microbial communities in immobilizing DOM in the form of refractory
substances over short timescales. The findings of this study enhance our understanding of the mechanisms
underlying the Patagonian Shelf's role as a hotspot for carbon sequestration (Kahl et al., 2017). The described
carbon route partially explains why the region acts as a carbon sink throughout most of the year, irrespective of
the primary productivity magnitude. These results underscore the critical role of the microbial carbon pump
(Jiao et al., 2024) as a key carbon sequestration pathway within non-frontal areas of the Patagonian Shelf. While
these findings highlight its importance, the seasonal relevance of this mechanism remains to be explored.
The accumulation of DOM was not limited to stations experiencing active phytoplankton growth, indicating
that net DOM release is not necessarily tied to specific phenological stages, as previously suggested (Bachi et
al., 2023). Moreover, it was not correlated with phytoplankton species composition, implying a low level of
specialization in the bacterial utilization of species-specific DOM substances, a trait that becomes apparent
under conditions of high resource availability (Sarmento et al., 2016). Instead, the degree of complexity of
DOM expressed as the prevalence of humic compounds ($FDOM_C$, $FDOM_A$, and $FDOM_M$), emerged as a





primary factor determining its utilization by bacteria. Detectable consumption occurred when the initial DOM
pool presented a high contribution of low reactivity compounds, likely resulting from the phenological status
of the sampling point as depicted by the temporal progression of satellite chlorophyll-a. This scenario was
observed at stations 23 and 21, where sampling was preceded by low chlorophyll-a content, indicating limited
autochthonous production of fresh DOM in the preceding month. However, the differences in bacterial growth
and FDOM-inferred biodegradation at both stations suggest variations in bacterial activity. At station 23, intense
bacterial activity was evidenced by the significant increase in the complexity of DOM and the net DOM
consumption under both treatments. Despite the high contribution of refractory compounds in this site ($FDOM_C$,
$FDOM_A$, and $FDOM_M$), the initial labile DOM pool was also high (particularly $FDOM_T$), denoting that bacteria
may have sustained their growth upon this labile initial stock over the 24 h incubation under the absence of
DOM-producing phytoplankton. Interestingly, under the absence of protists, DOM consumption shifted to
accumulation in station 21. At this station, a notable abundance of initial refractory compounds was observed.
In contrast to observations at station 23, bacterial activity was low, as indicated by a low growth rate and no
detectable biodegradation of DOM. Therefore, the apparent decrease of bacterial DOM utilization under the
absence of protists, suggests that bacterial reliance on phytoplankton DOM exudates increases when the existing
DOM pool consists primarily of refractory compounds. Despite similar phenological stage, this situation was
not observed in station 22, where the degree on refractory compound was lower. This suggests the presence of
other sources of labile DOM that may not be adequately represented by total chlorophyll-a measurements, such
as *Synechococcus*, a significant picophytoplankton genus in the shelf area (Silva et al., 2009) known to
contribute to bioavailable DOM production (Zheng et al., 2019).
The opposite scenario, i.e., a shift from weak DOM accumulation to DOM consumption under the absence of
protists, was observed in station 13. Here, the initial proportion of labile DOM was higher, denoting that DOM
bioavailability did not restrict bacterial activity. This station displayed the lowest biomass ratio between
phytoplankton and bacteria, indicating that DOM consumption in the absence of protists occurred when the
initial phytoplankton biomass significantly surpassed that of heterotrophic bacteria. This suggests that the initial
phytoplankton biomass masked intense bacterial DOM utilization, which only became evident upon the removal
of protists. However, DOM accumulation might be overestimated due to the input of DOM lysates resulting
from viral lysis. Indeed, the decrease in HIX alongside the increase in BIX observed in station 13 under both
treatments suggests that viruses are transferring bacterial biomass into the labile DOM pool (Proctor and
Fuhrman, 1991). The low bacterial growth and grazing at this site along with the station's bloom timing further
support this notion, as it suggests that sampling occurred at the transition between grazing control to viral
control of bacterial biomass. Indeed, the latter tends to occur under low grazing pressure (Bettarel et al., 2004;
Pasulka et al., 2015). Overall, the fact that DOM accumulation occurred even in the absence of protists, indicates
that bacteria are the main source of DOM as previously noted (Gruber et al., 2006). The initial proportion of
refractory compounds better predicted the net DOM production by providing insights on microbial succession
trajectories.





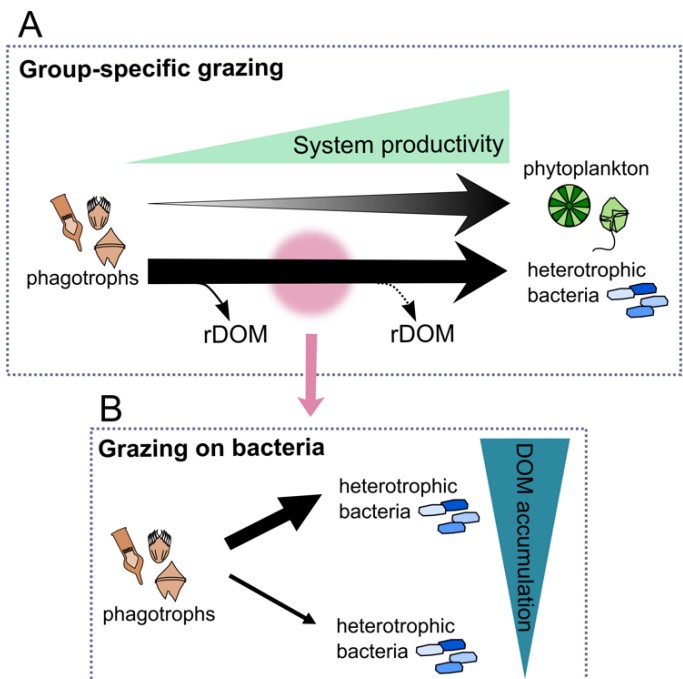


**Figure 9. Grazing-mediated pathways of DOM in the Patagonian Shelf. A. While phytoplankton biomass
was found to be twice that of bacteria, a selective grazing on bacteria was observe compensating for their
fast growth rate along the productivity gradient. In turn, grazing on phytoplankton increased with rising
productivity, suggesting that phytoplankton, particularly smaller cells, are increasingly grazed as a
byproduct of intense grazing on bacteria, aligning with the enhanced microbial loop hypothesis.
Heterotrophic bacteria were the primary agents shaping DOM quality, capable of storing carbon in the
refractory DOM pool (rDOM) even in the absence of DOM-producing protists. However, the production
of rDOM was more conspicuous in mid-shelf stations than in the shelf break. B. Grazing on bacteria
influenced the net production of DOM. Arrows thickness represents grazing pressure. Pronounced
bacterial grazing led to an accumulation of DOM, likely by reducing the biomass of bacterial standing
stock and by contributing with egestion organic substances.**

592

DOM also tended to accumulate under high bacterial mortality due to grazing. Our results evidenced that carbon
was primarily channeled from prokaryotes to protistan grazers, bypassing slower growing phytoplankton. This
group-specific grazing mortality align well with the grazing selectivity model, wherein grazers exhibit
preferences against high-growth-rate organisms, establishing a tight coupling between growth advantages and
grazing vulnerability across environmental gradients (Landry et al., 2023). Our experimental data not only
support this hypothesis but also provide new insights into the repercussions of bacterial grazing on dissolved
carbon stocks (Fig. 9b). Specifically, our results suggest that grazing on bacteria can lead to an accumulation
of DOM produced by phytoplankton by reducing the biomass of bacterial standing stock. In addition, protistan





grazing may contribute to the DOM pool by releasing bacterial carbon (Taylor et al., 1985). However, bacteria may not immediately utilize this DOM source, as adapting their enzymatic machinery to target new compounds requires additional energy expenditure, resulting in less efficient resource utilization (Baña et al., 2014). Similar results were observed in polar waters, where high protistan bacterivory was associated with DOM accumulation (Lund Paulsen et al., 2019). Our observations carry biogeochemical implications, as intense bacterial grazing implies that bacterial biomass becomes available to higher trophic levels, thereby circumventing the DOM cycle. In other words, while most bacterial biomass is directed by protistan grazers toward higher trophic levels, it also partially diverts the production of DOM lysates by viral lysis (Suttle, 2005). Our experimental results also indicate that intense grazing only partially compensates for this carbon route, as a fraction of phytoplankton-derived DOM remains unexploited by bacteria over short timescales.

The fate of grazing-derived DOM remains uncertain in our experimental setup, as it could either serve as a potential source for bacterial utilization, thus establishing a positive longer-term predator-prey feedback not captured in our 24-h experiment, or contribute to the complex DOM pool, feeding into the refractory fraction. Indeed, previous observations revealed that DOM derived from protistan grazers varies in its bioavailability (Taylor et al., 1985), and their complexity is further shaped by taxa composition (Gruber et al., 2006; Nagata and Kirchman, 1992). In our experiments, we did not observe a clear trend in DOM transformation between treatments with and without protists, indicating that bacteria remain as the primarily shapers of DOM quality. Overall, our finding revealed that under high bacterial growth rate that follows the onset of the productive season, protistan grazers not only channels carbon trough remineralization but also, foster the degree of DOM accumulation by reducing DOM-degrading bacterial stock and contributing with egestion substances. Additionally, instead of acting solely as a sink of carbon trough mineralization of organic compound, bacteria serve as a crucial link between assimilatory $CO_2$ and higher trophic levels.

**Data availability**

DOI: 10.5281/zenodo.11662261

**Authors contribution**

CLA conceptualized, designed, and carried out the experiments, analyzed plankton samples and acquired funding. JEGC, AM and ASG analyzed DOC, CDOM, FDOM and nutrients samples and interpreted the results. JEGC performed the PARAFAC multivariate algorithm and calculated fluorescence indices. RS analyzed plankton samples and performed biomass calculations. JCM contributed to the numerical methodology design and the conceptualization of overreaching goals. LARE analyzed and interpreted CTD data and Moderate Resolution Imaging Spectroradiometer (MODIS) Aqua images of chlorophyll-a. RL coordinated responsibilities for the research activities planning and execution, acquired funding and contributed to the conceptualization of overreaching goals. CLA prepared the manuscript with contributions from all co-authors.



**Competing interests**
The authors declare no competing interests.

**Acknowledgments**
We are thankful to the crew of the RV "Dr. Bernardo Houssay" of Prefectura Naval Argentina for their support
on field activities and sampling. This study was supported by the National Agency for Promotion of Science
and Technology (FONCYT-PICT 0467–2010 and FONCYT-PICT 2386-2017), the National Scientific and
Technical Research Council (CONICET-PIP 11220200102681CO) and by the Argentine Oceanographic
Institute (IADO, CONICET-UNS). The authors acknowledge that language checking and spelling
improvements were done by ChatGPT (powered by OpenAI's language model, GPT-3; http://openai.com).

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
