# Peer review of "The bacteria-protist link as a main route of dissolved organic"

_EGUsphere, 2024_

## Author Response (AR1)

**RC 1**

This study presents incubation experiments conducted on the Patagonian Shelf to investigate DOM consumption and accumulation. The authors found that the food web at productive stations is primarily based on phytoplankton and heterotrophic bacteria, while in less productive regions, the food web showed an inverted pyramid structure due to rapid bacterial growth rates. Key findings suggest that intense grazing on heterotrophic bacteria promotes DOM accumulation, and that DOM consumption is influenced by FDOM properties. While the topic is interesting, I cannot accept this manuscript in its current form. The methods and results are not robust enough to support the conclusions. The manuscript is poorly organized, and the discussion lacks clarity. The writing needs significant improvement, as many statements are redundant and difficult to follow.

Authors reply: Thank you for your feedback. In the revised version of the manuscript, we have made the presentation more concise, significantly reducing the word count (~6500 words without references). We have also carefully reorganized the methodology and discussion sections to improve readability.

Below are my major concerns:
1. The determination of bacterial and phytoplankton growth and grazing rates is unclear and unconvincing. The potential effects of nutrient addition and viral lysis on the results need to be addressed. The authors should verify the validity of their methods and discuss how these factors might have influenced the results.

Authors reply: The dilution method is a widely accepted technique since it was first introduced by Landry and Hassett in 1982 (Marine Biology). Comparisons between the growth rate derived from dilution experiments have revealed a high correlation (close to 1 at rates ≥0.5 $d^{-1}$, and 0.75 at rates ≤0.4 $d^{-1}$) with rates obtained from the 14C approach (Laws 2000 *Deep Sea Research II*, Calbet and Landry 2004, *Limnology and Oceanography*). In our study, the measurement of bacterial growth and grazing adheres to the same theoretical framework and assumptions as the original protocol, which has been consistently applied since its release (e.g., Tremaine and Mills 1987, *Applied Environmental Microbiology*).

We acknowledge that the effectiveness of the dilution method is contingent upon specific assumptions that should be considered: (i) phytoplankton growth rates are not density-dependent, that is, they are not influenced by dilution, (ii) nutrient levels and light conditions are maintained consistently across all dilutions, thereby preventing them from becoming limiting factors for phytoplankton growth, (iii) phytoplankton consumption is linearly related to prey abundance, (iv) consumer density remains constant during the incubation period, and (v) phytoplankton growth can be accurately modeled using an exponential growth equation.

To mitigate the limitations of the method and to enhance the consistency of our assessments, we supplemented the incubation bottles with excess nutrients according to the Redfield ratio, while also including control treatments without added nutrients to gain reliability of growth rate estimates.

However, we do not exclude the possibility that viral lysis may influence growth estimates (Staniewski and Short 2014, *Aquatic Microbial Ecology*). This potential effect is discussed in lines 501-504, as detailed below:

"While protistan grazing was the main factor for bacterial and phytoplankton mortality, viral lysis likely played an unrecognized role, possibly enhancing prey availability for protists by releasing cellular lysates or reducing competition (Staniewski and Short, 2014). This dual impact warrants further exploration to fully understand microbial influences on DOM dynamics."

2.      I disagree with the use of a254 to indicate net DOM production. This parameter merely reflects absorbance at 254 nm and does not directly measure DOM production. Authors reply: Our use of this proxy was informed by previous research indicating that absorbance at 254 nm is strongly correlated with CDOM (e.g., Brandstetter et al. 1996, *Zeitschrift für Pflanzenernährung und Bodenkunde*). However, we agree with the reviewer that the terminology in the original manuscript could be refined to better convey that this parameter may not comprehensively represent all DOM components. To address this, we have revised the text and figures to clarify these limitations and ensure precise communication of the method's scope and constraints.

3.      The explanation regarding protist grazing on bacteria rather than phytoplankton is insufficient. Differences in the protist community are also significant drivers of this process. The discussion on how productivity influences food web structure is lacking. Additionally, the low phytoplankton growth rate needs a more compelling explanation.

Authors reply:
"The explanation regarding protist grazing on bacteria rather than phytoplankton is insufficient. Differences in the protist community are also significant drivers of this process."
We agree that the structure of the protist community is an important driver of grazing selectivity. We have added this text regarding the potential role of consumers composition on grazing (lines 449-455):

"An additional factor contributing to grazing selectivity across productivity areas may be related to grazer community composition, which differed between shelf and upwelling areas, with ciliates more concentrated on the shelf and heterotrophic dinoflagellates more associated with upwelling stations (especially in stations 12 and 13). In contrast, phytoplankton structure showed no distinct regional differences. Ciliates primarily select prey based on size due to their oral diameter, while dinoflagellates employ diverse feeding strategies, allowing them access to a broader range of prey sizes. This trait positions dinoflagellates as key grazers in meso- and eutrophic environments, such as upwelling areas (e.g., Sherr and Sherr 2007; Calbet 2008; Hansen et al. 1997)."

"The discussion on how productivity influences food web structure is lacking."
Thank you for this suggestion, we added this text into the new version of the manuscript (lines 412-424):

"In oligotrophic areas, microbial food webs involve multi-step carbon transfer dominated by small organisms that efficiently recycle nutrients. By contrast, productive regions have shorter, more efficient food webs where carbon moves directly to higher trophic levels (Armengol et al., 2019). Our observations showed stronger microbial trophic coupling on the mid-shelf compared to the upwelling zone, where most primary producers escaped grazing. Mid-shelf stations, including post-bloom station 12, exhibited an "inverted pyramid" structure with consumer biomass exceeding that of prey, suggesting high trophic efficiency (McCauley et al., 2018). In the upwelling zone, despite evidence of protistan herbivory, much of the phytoplankton remained ungrazed likely due to strong top-down pressure from microcrustaceans on both phytoplankton and protistan grazers. This created a more typical bottom-heavy biomass structure, indicative of high primary productivity where some primary production bypasses protist predation and is consumed by microcrustaceans or exported. These observations align with previous findings in the shelf region, showing strong trophic coupling and minimal sinking of unused biomass, dominated by small phytoplankton (<5 µm) and balanced biomass levels of micro- and mesozooplankton (Negri et al., 2013)."

"Additionally, the low phytoplankton growth rate needs a more compelling explanation."
Authors reply: Low phytoplankton growth may be related to bloom timing, as low growth rate of phytoplankton was detected under bloom initiation phases in stations 21 and 14. According to the temporal progression of satellite chlorophyll-a the days before and after the sampling took place, the rest of the stations were at a post-bloom or stationary phase, where net growth rate is expected to be close to zero. An additional factor, not discussed in the original version of the manuscript, is that some eukaryotic pigmented plankton may suffer from UV photoinhibition when exposed to deck conditions. Although light conditions were carefully controlled during handling, and the incubator was covered with a special net to prevent light overexposure, we acknowledge that continuous light exposure does not replicate the natural mitigation of UV stress provided by vertical mixing in the water column. This is primarily due to the difficulty of replicating site-specific UV penetration and the mixing conditions that determine the residence time at various depths. We added some text acknowledging this issue in the new version of the manuscript (lines 456-464):
"Regarding the low phytoplankton growth rates observed in this study, an additional factor beyond bloom timing may involve UV photoinhibition, which can affect some eukaryotic pigmented plankton when exposed to deck conditions. Although light conditions were carefully controlled during handling and the incubator was covered with a special net to prevent overexposure, continuous light exposure cannot fully

replicate the natural mitigation of UV stress provided by vertical mixing in the water column (Häder et al., 2015). Laws et al. (2000) noted unrealistic growth estimates, attributing the negative effects of UV exposure to phytoplankton incubated incubation bottles at the surface. Simulating natural UV exposure remains a common challenge in most incubation experiments, where communities experience more stable light conditions."

1. The manuscript confuses CDOM and FDOM. While FDOM is a subset of CDOM, the manuscript treats them as independent terms without explanation. The claim that DOM consumption is influenced by FDOM properties lacks direct evidence. While I agree that DOM consumption is influenced by the initial labile fraction, this pertains to DOM quality, not FDOM properties. If FDOM impacts DOM consumption, the manuscript should specify which FDOM components are involved and how they affect the process.

Authors reply: We agree with the reviewer that there was an unclear differentiation between CDOM and FDOM in the original version of the manuscript. For this reason, we clarified that from the chromophoric fraction we only measured the absorbance at 254 nm, which provides an estimation of the DOM amount (e.g., Brandstetter et al. 1996, *Zeitschrift für Pflanzenernährung und Bodenkunde*) We removed the abbreviation CDOM from the entire manuscript and reformulated the corresponding passages accordingly.
We also clarified that the different FDOM components allowed us to track the possible sources and changes in the diagenetic state of DOM, i.e., its production and transformation.

There are also numerous formatting errors regarding abbreviations and citations. Please carefully review and correct these throughout the manuscript.
Authors reply: We carefully reviewed and corrected any inaccuracies in the text, including abbreviations and citations, to ensure clarity and precision throughout the manuscript.

Specific comments
Abstract: The abstract should be more concise. Summarize the main results and provide compelling explanations.
Authors reply: We rewrote the Abstract to make it more concise and compelling.

Lines 36-37: The phrasing is awkward and difficult to follow.
Authors reply: This sentences was reformulated as follows:

"High trophic efficiency contributed to a top-heavy biomass distribution, where consumer biomass exceeded that of producers. However, in highly productive stations influenced by nutrient-rich upwelling waters, a traditional pyramid structure was observed."

Introduction: The rationale for conducting this study is unclear. I suggest starting with the significance of DOM, followed by the biological processes influencing its production and consumption. Then, provide background information on CDOM and the sampling region.
We reformulated the introduction section according to the reviewer's suggestion.

Line 48: The phrase "consensual impact" is awkward.
Authors reply: This sentence has now changed. The term "consensual impact" was removed.

Line 50: Spell out "CO2" when it first appears. Please ensure this is consistent throughout the manuscript.
Authors reply: Done.

Line 69: Avoid starting a paragraph with an abbreviation.
Authors reply: Done.

Lines 75-77: Rephrase this statement for clarity.
Authors reply: Done. The sentence now reads as follows (lines 52-54):

"Protistan grazers, in particular, not only recycle nutrients and enhance bloom sustainability but also impact carbon cycling through DOM production via biomass reworking and excretion (Kujawinski et al., 2004; Baña et al., 2014)."

Line 98: bacteria are producer?
Authors reply: Thank you for this observation. Bacteria are not producers in the strict sense of carbon production; instead, they play a role as decomposers or recyclers in the ecosystem. In the original text, the term was used to contrast with consumers, which may have led to confusion. We have revised the sentence to ensure clarity and avoid any misinterpretation.

Materials and Methods: This section needs reorganization. Currently, it lacks a clear logical flow and is overly lengthy. Focus on the incubation experiments and clearly describe how subsamples were collected and for what parameters. Then, detail the methods for each parameter analysis. More details are needed on the quantification of phytoplankton and bacterial growth and grazing rates. As mentioned earlier, using a254 as an indicator of net DOM production is not convincing.
Authors reply: We have reworked the Methodology Section to make it more concise and focused on the description of incubation experiments according to the reviewer's suggestion.

Line 124: "l" should be capitalized to "L".
Authors reply: Done.

Line 127: A comma is missing.
Authors reply: Done.

Line 147: Explain what these percentages represent.
Authors reply: Done.

Line 154: Specify the amount of nutrients added to the incubation experiments.
Authors reply: Done.

Line 155: Is the unit of Chl-a "µm L$^{-1}$"?
Authors reply: Thank you for this remark, the Chl-a units are µg L$^{-1}$. We corrected the misspelling.

Line 179: You mention sampling pico- and nanoplankton, but I did not find an analysis of picoplankton.
Authors reply: Thank you for pointing this out. Picoplankton was analyzed using the same methods employed for nanoplankton, incorporating both light and epi-fluorescent microscopy. We acknowledge inconsistencies in the terminology used to describe plankton size fractions in the original manuscript, which were also noted by another reviewer. To address this, we have standardized the terminology throughout the manuscript to accurately represent these groups. Specifically, we now refer to the smaller size range of nanoplankton by explicitly indicating the 2-5 µm size class, rather than using the term "picoplankton".

Line 193: Clarify whether "ultraplankton > 5µm" is correct.
Authors reply: In line with the previous reply, this term was removed from the manuscript and now we refer to the size range of nanoplankton by explicitly indicating the 2-5 µm and the 5-20 µm size class, rather than using the term "picoplankton".

Lines 233: Add a space between "220" and "nm", as well as between "370" and "nm".
Authors reply: Done.

Lines 241-242: Explain how the indexes were derived and how they indicate the corresponding processes.
Authors reply: We added this text following the reviewer's suggestion (lines 177-184):

"Humic-like material was represented by three fluorophores: FDOMC (Ex/Em: 350/440) and FDOMA (Ex/Em: 250/425) of terrestrial origin, and FDOMM (Ex/Em: 310/380) of marine origin. FDOMC indicates highly unsaturated components, FDOMA reflects a moderate degree of unsaturation, and FDOMM suggests a low degree of unsaturation. Protein-like material was characterized by two fluorophores: FDOMB (Ex/Em: 260/300) and FDOMT (Ex/Em: 270/330), both derived from autochthonous biotic sources. FDOMB is associated with compounds similar to tryptophan and

tyrosine, linked to recent organic matter production through primary productivity, while FDOMT indicates tryptophan-like compounds formed from microbial protein breakdown (Jørgensen et al., 2011; Drozdova et al., 2022)."

Discussion:
Also refer to my major concerns
Line 438: Delete parentheses
Authors reply: Done.

Line 475-476: "since the turnover...", how did you get to this statement?
Authors reply: This sentence has now changed.

Line 534: Cite the earlier paper by Jiao et al. 2010.
Authors reply: Done.

Line 575: In Line 563, you state that DOM accumulation shifts to DOM consumption in the absence of protists. This needs clarification.
Authors reply: Thank you for pointing out this potential contradiction. In Line 575, we intended to convey that DOM accumulation can occur even without the presence of protists, though this is not always the case. We agree that the original phrasing could be confusing, so we have revised these sentences for clarity and consistency to avoid any misinterpretation.

Line 577: change "on" to "into"
Authors reply: Done.

Figures:
the abbreviation and error bar shown in the figures should be explained in caption.
Authors reply: We have thoroughly reviewed and corrected any inaccuracies in the figures and figure's references to ensure no missing elements and precision throughout the manuscript.

**RC 2**

López-Abbate et al investigated how phytoplankton and bacterial mortality influence the pool of dissolved organic matter (DOM) on the Patagonian Shelf; a region of the ocean known as a hotspot of carbon sequestration. Dilution experiments revealed that phytoplankton mortality via grazing was greater than their growth rate, while bacterial growth and mortality were relatively balanced. By measuring colored dissolved organic matter (CDOM) and fluorescent dissolved organic matter (fDOM), authors observed a decrease in 'fresh' DOM and an increase in aromatic DOM. How carbon flows through multiple trophic levels is of importance for understanding marine ecosystems, especially in the context of carbon dioxide sequestration. While this is valuable environmental data, the current state of the manuscript is not ready for publication. Besides some grammatical, spelling, and nomenclature errors, figures were unclear, data was not always appropriately discussed, and some of the data does not support the conclusions drawn by the authors. Here are my major concerns:

1. DOM production and consumption as inferred by absorbance at 254 nm is likely not a good proxy for the total DOM, but only a subset of DOM. This becomes especially problematic when the authors at line 521 claim there was a general trend denoting weak DOM limitation of bacterial growth because organic matter accumulated – this result does not suggest limitation. It is known that heterotrophic bacterial consumption of labile organic matter generates refractory organic matter that would accumulate (see Lechtenfeld et al 2015, nature comms).

Authors reply: We appreciate the reviewer's insightful comment regarding the distinction between absorbance at 254 nm and the total amount of DOM. To address this, we clarified in the revised manuscript that absorbance at 254 nm represents the chromophoric fraction of DOM and serves as an estimation of the total DOM amount (e.g., Brandstetter et al. 1996, *Zeitschrift für Pflanzenernährung und Bodenkunde*). We have also emphasized that this proxy is primarily intended to facilitate comparisons between initial and final experimental conditions, as well as across treatments, rather than providing a precise quantitative estimate of the total DOM pool. Furthermore, we acknowledge that net changes in absorbance at 254 nm, as an indicator of DOM accumulation, do not necessarily preclude the presence of bacterial carbon limitation. We have revised the original text to make these points clearer.

2. Section 4.2, while protists were removed from the incubations by GFF and bacteria were retained in the filtrate, some portion of these are likely to be cyanobacteria that would skew the interpretation of DOM accumulation or depletion due to heterotrophic bacterial activity.

Authors reply: We agree with this remmark. In particular, we acknowledge that attributing DOM sources and consumption solely to heterotrophic bacteria is inaccurate, as it overlooks the potential contribution of cyanobacteria and viruses. Thank you for your feedback. We addressed this limitation in the Discussion section as follows (lines 504-510):

"Furthermore, most changes in DOM production and consumption in protist-free bottles were attributed to heterotrophic bacteria. However, filtration likely allowed some cyanobacteria and viruses to pass through, contributing additional DOM sources. Synechococcus, the dominant cyanobacterium in the Patagonian Shelf, releases bioavailable DOM that stimulates specific bacterial activity (Wang et al. 2022). Viral lysis also converts microbial biomass into DOM (Chen et al. 2022). These overlooked sources likely influenced the short-term fate of DOM and contributed significantly to the DOM in the incubation bottles."

3.      Section 3.1 and following discussions of different phenological stages regarding a 'phytoplankton bloom': Please refer to Behrenfeld & Boss (2017, Global Change Biology) for a quantitative definition of phytoplankton blooms. Based on the paper above, the concentration of phytoplankton biomass to determine a blooming state is qualitative. Phytoplankton concentration can change purely from changes in mixed layer depth. This qualitative definition does not set the stage for a good explanation of the observations in this paper as I am not convinced that the current interpretations of phenological states are correct.

Authors reply: We acknowledge that factors such as wind stress, topographic anomalies, and mixed layer depth may contribute to plankton aggregation alongside the seasonal development of productive events. However, the phytoplankton phenology of the Patagonian Shelf and Shelf Break has been extensively documented. The recurrence of the spring phytoplankton bloom has been well established through satellite data (e.g., Romero et al. 2006, *Journal of Geophysical Research*; García et al. 2008, *Deep Sea Research Part I: Oceanographic Research Papers*; Delgado et al. 2023, JGR: Oceans) as well as through direct observations (e.g., Lutz and Carreto 1991, *Continental Shelf Research*, Ferronato et al. 2021, *Food Webs*; Guinder et al. 2024, *Springer Nature*). In addition to the consistent nutrient supply provided by the Malvinas Current, which enriches both the frontal area and inshore surface waters, the seasonal phytoplankton bloom is supported by the seasonal stratification of the water column. This stabilization of the water column progresses latitudinally, following the thermal cycle. In the latitudinal range of our study, the bloom typically begins in early spring (September and October) (Rivas et al. 2006 *Continental Shelf Research*, Romero et al. 2006, *Journal of Geophysical Research*).

Despite having robust information regarding the recurrent nature of the spring phytoplankton bloom in the Patagonian Shelf, we acknowledge the reviewer's point that a clear definition of a bloom is lacking. Providing this definition would help readers better understand how we, the authors, interpret a blooming event in this region of the world ocean. We added this text to the Methodology Section based on the Behrenfeld & Boss student's tutorial (lines 108-111):

"We define blooming conditions as periods where the net rate of biomass change (specific division rate minus specific loss rate) remains positive long enough to result

in net biomass accumulation (Behrenfeld and Boss 2017). Factors like wind stress that may influence plankton aggregation are not considered contributors to these conditions.."

4.      Section 3.4: I am unclear why there was no measure of phytoplankton growth rates in this experiment. Even if they are low they should be reported, and if they are not growing for some reason, this needs to be elaborated on in the discussion, especially why the nutrient amended treatments did not grow

Authors reply: Phytoplankton and bacterial growth rates ($\mu$, $d^{-1}$) were measured in all experiments. Following the dilution method (Landry and Hassett 1982, *Deep Sea Research II*), linear regressions of net growth rate versus the dilution factor under nutrient-enriched conditions were used to estimate phytoplankton growth ($\mu$, indicated by the y-intercept) and mortality from protist grazing (m, represented by the slope of the regression). For experiments where negative y-intercept values were observed, these values were adjusted to 0.01 $d^{-1}$ to prevent division by zero when calculating the proportion of primary or bacterial production consumed by grazers (m:$\mu$). Similarly, negative or non-significant slope values for protistan grazing rates (m, $d^{-1}$) were set to zero, as suggested by Calbet and Landry (2004, *Limnology and Oceanography*).

Negative y-intercepts were detected in four out of six phytoplankton growth rate estimates (stations 12, 13, 22, and 23), while non-significant slope values influenced estimates at mid-shelf stations (stations 21, 22, and 23). In response to the reviewer's helpful feedback, we have included additional clarifications in the methodology section. Moreover, we acknowledge that the discussion regarding low phytoplankton growth estimates needed further elaboration. This was also highlighted by another reviewer, and we have now expanded the discussion section to provide a more comprehensive explanation (lines 456-464):

"Regarding the low phytoplankton growth rates observed in this study, an additional factor beyond bloom timing may involve UV photoinhibition, which can affect some eukaryotic pigmented plankton when exposed to deck conditions. Although light conditions were carefully controlled during handling and the incubator was covered with a special net to prevent overexposure, continuous light exposure cannot fully replicate the natural mitigation of UV stress provided by vertical mixing in the water column (Häder et al., 2015). Laws et al. (2000) noted unrealistic growth estimates, attributing the negative effects of UV exposure to phytoplankton incubated incubation bottles at the surface. Simulating natural UV exposure remains a common challenge in most incubation experiments, where communities experience more stable light conditions."

5.      Viral mortality was not measured or estimated and thus should be removed from the abstract. This study does not compare the relative contribution of DOM from viral mortality relative to grazing mortality

Authors reply: We agree with this remmark, we removed all references to viral activity from the abstract. A short comment regarding the potential role of viruses was included in the Discussion Section (lines 501-504):

"While protistan grazing was the main factor for bacterial and phytoplankton mortality, viral lysis likely played an unrecognized role, possibly enhancing prey availability for protists by releasing cellular lysates or reducing competition (Staniewski and Short, 2014). This dual impact warrants further exploration to fully understand microbial influences on DOM dynamics."

6.      Section 3.2: There is no discussion of nitrite, which is makes up a good portion of total nutrients at most stations. This section has multiple errors that need attention. For example, line 288 says station 14 had the highest nutrient concentration, but based on figure 2 this is station 12. Additionally, there is no A or B in figure 2, which is referenced in text.
Authors reply: We appreciate the reviewer's attention to these details. We realize that there was an error in the figure legend where we mistakenly labeled nitrate as nitrite and vice versa. We apologize for this oversight. We corrected the figure and its legend in the revised version of the manuscript to accurately represent the data. We have also thoroughly reviewed and corrected any inaccuracies in the text, figures, and figure's references to ensure clarity and precision throughout the manuscript. Assuming that the reviewer was referring to nitrate and not nitrite, we have also added a comment on nitrate in the Discussion Section as follows (lines 392-398):

"The contrast in chlorophyll-a concentration between shelf and upwelling areas may be related to nitrate content, which is the main limiting nutrient on the Patagonian Shelf (Paparazzo et al. 2010). The Shelf Break receives a continuous supply from the nutrient-rich upwelling of the Malvinas Current, while nitrate on the Shelf is mainly provided from the northward flow of Subantarctic Waters. However, as it moves north, phytoplankton uptake depletes nitrate, leading to lower concentrations on the northern Patagonian Shelf (Song et al. 2016). This creates a regional contrast in nutrient levels between the Shelf and the Shelf Break."

7.      Throughout the manuscript the terms statistically significant and significant are used. It is hard to tell if a phrase like 'Significant grazing effect on bacteria was found in all experiments' on line 362 is a statistically significant result or was just a pronounced effect. Please refrain clarify when things are statistically significant, and use another word for when there were strong effects
Authors reply: We reviewed the use of the term 'significant' throughout the manuscript and avoided applying it in contexts not supported by statistical tests. For growth and grazing estimates, 'significance' was only used in reference to linear regression analysis (Landry and Hassett 1982). In this analysis, the null hypothesis states that no

linear relationship exists between the variables, which is tested against the alternative hypothesis that the slope differs from zero. Thus, growth and grazing rates were only considered significant when the hypothesis test for linearity between the apparent growth rate and the dilution factor yielded a p-value < 0.05.

Minor comments:
Line 48: The term 'consensual' implies humans and microbes agreed to impact each other...this is not the correct term to use
Authors reply: This text has now changed. We remove the term "consensual impact".

Line 154 – 156: confusing sentence, what concentration of inorganic nutrients were added? I don't understand how this has to do with assuming the maximum chlorophyll-a concentration. The next line says that the amount of nutrients was taken from another paper which may be why the phytoplankton in these amended conditions did not grow much.
Authors reply: Nutrient amendment is a standard practice in grazing experiments to fulfill one of the key assumptions of the dilution technique: that nutrient levels remain uniform across all dilutions to prevent nutrient limitation of phytoplankton growth. However, nutrient addition can artificially enhance phytoplankton growth rates, potentially leading to results that may not accurately represent natural conditions (Landry 1993, *Handbook of Methods in Aquatic Microbial Ecology, Lewis Publishers*). To address this, a refined method based on the Redfield ratio has been suggested, which tailors nutrient addition to the phytoplankton growth rate and expected chlorophyll-a concentration specific to the time and location of the experiment (Calbet and Saiz 2018, *Journal of Plankton Research*). In our study, the concentration of nutrients added was calculated using available regional literature (e.g., Delgado et al. 2023, *JGR: Oceans*). To ensure a more accurate representation of natural growth conditions, we also included control treatments without nutrient amendments. For clarity, we have now provided the nutrient concentrations used in the incubation bottles for both the Mid-shelf and Shelf Break stations (lines 124-125).

Line 205-206, and figure 5: Phytoplankton functional group is not well-defined, and to me, the term refers to the function of the organism (e.g. nitrogen fixers, calcifiers, etc.), not the general category. Is there a citation for this terminology?
Authors reply: Thank you for pointing out this issue. We acknowledge that our classification of plankton groups may have been unclear. We did not base our categorization on functional roles (as described by e.g. Mitra et al. 2016, *Protist*). Instead, we grouped plankton by their primary mode of energy acquisition (phagotrophic vs. phototrophic) and further distinguished them based on size. We added the following text to clarify this categorization and revised the manuscript to ensure that the distinction between these groups is accurately represented (lines 156-160):

"Protistan taxa abundance was visualized by a heatmap (employing the R package heatmaply). Heterotrophic and autotrophic taxa were segmented into groups based on size, distinguishing between nanoplankton (2-20 μm) and microplankton (20-200 μm) (Sieburth et al., 1978). Nanoplankton included heterotrophic nanoplankton (HNP), phototrophic dinoflagellates (PD), coccolithophores, and phototrophic nanoplankton (PNP), while the microplankton comprised ciliates, heterotrophic dinoflagellates (HD), and diatoms."

Please use defined size classes (see Sieburth 1978). I had not heard the term 'ultraplankton' before and in trying to look through the literature I see multiple definitions for this term. As defined in text this says >5um, which would be inclusive of nanoplankton and everything else <200um

Authors reply: We revised the entire manuscript to clarify the terminology and ensure that the distinction between these groups is accurately represented. We removed the term "ultraplankton" and referred to the lower size range of nanoplankton by explicitly mentioning the size class, i.e., plankton in the 2-5 μm size range.

Figure 4A it is unclear why Phyto and HB are grouped together, while phyto and phagotrophs are separate, and what these pyramids represent. Abbreviations need to be defined in the figure legend.

Authors reply: This figure's panel aims to provide a rapid visualization of the food web structure regarding the contribution of biomass of both consumers (phagotrophs) and their prey (phytoplankton and heterotrophic bacteria). Phytoplankton and heterotrophic bacteria are summed together in order to identify phytoplankton plus heterotrophic bacteria–based food web type according to the methodology described by Kang et al. (2023, *Science Advances*). We add the following text to the figure's legend to improve clarity:

"Figure 4.A. Polygons represent the type of food web structure based on the ranking of carbon biomass of phagotrophs, phytoplankton (phyto), and heterotrophic bacteria (HB). Upper numbers indicate station number. Violet polygons indicate inverted or top-heavy pyramids, where the biomass of consumers (phagotrophs) surpasses that of their prey (phytoplankton plus HB). In contrast, green polygons represent the more conventional bottom-heavy structure, characterized by greater prey biomass relative to that of consumers."

All figures in the new manuscript version were screened to include all missing elements.

In the introduction, please cite Moran et. al. 2022 (Limnology and Oceanography). This paper has a modern accounting of the movement of organic carbon through the surface ocean and would be of relevance to this study.

Authors reply: Done.

Figure 1A is missing a legend on the color bar
Authors reply: Thank you for your remark. We added the title of the color bar.

Figure 2:  Stylistically, I am not sure why in the figure DOC concentration is represented as a solid black line while nutrients are stacked bar plots.
Authors reply: The layout of this figure was designed to enhance clarity and effectively summarize the data while avoiding an excessive number of figures.

Figure 4C: y-axis limits should be changed so that growth rates and mortality can be seen. If they are truly zero for stations 23 and 22, then this needs to be elaborated on (see major comments)
Authors reply: Experiments that produced negative y-intercept values were adjusted to $\mu=0.01$ d$^{-1}$. The potential causes for low phytoplankton growth rates are given in a previous reply (major comment #4).

Figure 8: y-axis legends are missing units
Authors reply: Done.

Figure 9 is confusing – why are heterotrophic bacteria displayed twice in panel B? How are you defining system productivity in panel A? Especially given that measured phytoplankton growth rates were negligent along this gradient, I am unsure how productivity is being defined.
Authors reply: We created a new version of figure 9 according to the reviewer's suggestions.

Lines 618-622: through is misspelled as trough
Authors reply: We corrected the misspelling.

**RC 3**

This manuscript is addressing the effects of selective grazing of phytoplankton and bacteria on DOM. They are reporting an observation that a short-time DOM accumulation was accompanied by bacterial grazing.

Dilution experiments rely on linear regression analysis to estimate slopes and intercepts, and it's common to show dilution vs growth rate plots in dilution experiment studies, at least as supplementary figures. These plots give us information on statistical significance and errors associated with parameter estimates. This is particularly important in dilution experiments where biological rates often vary between replicates.

Authors reply: Thank you for your suggestion regarding the inclusion of dilution plots used to estimate growth and grazing rates. We understand the importance of visualizing these data; however, these plots are not typically included in published articles because the rate estimates are assessed for significance through regression analysis, which is detailed in the methods section. Including all dilution plots would substantially increase the length of the manuscript without providing additional meaningful insights, as the statistical rigor behind the regression analysis already ensures the reliability of the reported rates. Nonetheless, if the reviewer believes it would be beneficial, we can provide representative examples of these plots in the supplementary material to support transparency and clarity.

Figures 4C, 4D, and 8A show values without having errors associated with those data points, and readers cannot evaluate the results.

Authors reply: Figures 4C and 4D display growth and grazing rate estimates. Each dilution experiment involves three replicate bottles, from which chlorophyll-a concentrations and heterotrophic bacterial abundances are measured. These replicates are used to plot the apparent growth rates against the dilution factor, allowing us to calculate the slope (grazing rate) and y-intercept (growth rate). The resulting rate estimates are single values with no associated standard error, as each experiment yields a singular estimate rather than a distribution. The significance of these estimates is tested through regression analysis. Including error bars would therefore not accurately represent the data and could potentially mislead readers about the nature of the statistical analysis performed. Similarly, figure 8A is a regression plot and displayed data has no associated error.

I was expecting relevant information to be in "DOI: 10.5281/zenodo.11662261", but there is no such deposition in zenodo as far as I searched for it, and I couldn't address this point.

Authors reply: The data is currently under a one-year embargo period from the date of this article's submission, as the results have not yet been published elsewhere. However, we would be glad to share the data privately for the purpose of this review. Please let us know if you would like us to send the data via email at mclabbate@iado-conicet.gob.ar.

Although the authors state that the bacteria-protist link is the main determinant of DOM, the quantitative importance of this process is unclear throughout the manuscript. For example, what's the amount of DOC controlled by this process relative to the total DOC production in this system.

Authors reply: We appreciate the reviewer's concern regarding our methodology and the interpretation of our findings. Indeed, our statement was based not on direct DOM measurements during the incubation process, but rather on monitoring the absorbance at 254 nm. While absorbance at this wavelength serves as an estimate for the chromophoric fraction of DOM and can indicate total DOM levels (e.g., Brandstetter et al. 1996, *Zeitschrift für Pflanzenernährung und Bodenkunde*), we acknowledge that this approach may not comprehensively account for other organic substances. To address this, we have revised the original text to clarify this limitation.

Concerning the comparison between the contribution of DOC influenced by bacterial grazing and the background environmental DOC, we recognize that the current dataset does not allow for a conclusive analysis. Our experiments were not designed for quantitative DOM or DOC measurements but rather aimed at a comparative analysis between initial and final incubation conditions and between treatments, while the monitoring of FDOM was aimed at the examination of potential sources and diagenetic changes in DOM during the incubations.

I didn't think this manuscript is friendly to readers. It's lengthy. Information that should be in Introduction is often in Material and Methods. Abbreviations are often not explained in the first time they are used. Spaces are often missing in the text and figures.

Authors reply: Thank you for your feedback. In the revised version of the manuscript, we have made the presentation more concise, significantly reducing the word count. We have also carefully reviewed and corrected any inaccuracies in the text, including abbreviations and spacing, as well as in the figures and their references, to ensure clarity and precision throughout the manuscript.

Units are not consistent between parameters (L or m-3; M or mol/L).

Authors reply: We confirm that the unit "mol/L" was not used in the manuscript; concentrations of inorganic nutrients and DOC were consistently expressed in µM throughout. Additionally, we did not find any instances of the unit "$m^{-3}$" in the text. Following another reviewer's request, we have standardized the capitalization of "liter" to "L" throughout the manuscript. We have thoroughly reviewed and corrected any inaccuracies in the use of units to ensure clarity and consistency across the manuscript.

Using µm for µmol is confusing in the context of this study since µm is also for micrometer.

Authors reply: If the reviewer is referring to the use of "µM" in the manuscript, we would like to clarify that this is the standard and accepted notation for micromolar concentration.

---

## Author Response (AR2)

**ROUND 2**

**RC1**

The revised manuscript has been substantially improved. However, I still have some comments and suggestions that require further attention.

Concerns regarding the use of a254 to indicate DOM accumulation or consumption: The changes in a254 only reflect variations in the concentration of aromatic organic compounds, rather than the entire DOM pool. This limitation should be explicitly stated in the manuscript to avoid misinterpretation. I suggest the authors add the relevant clarifications in both the Introduction and Discussion sections.

Authors reply: We fully agree with your observation and to address this point, we have added a clarification in the introduction (P3, L79-80) and discussion (P18, L499-502) sections and expanded the methodology section with an explanation as follows:

"Absorbance at 254 nm represents 50–60% of the carbon content in marine DOM (Görs et al., 2007). It primarily embraces unsaturated compounds with aromatic rings, conjugated systems, or carbonyl groups, mostly of biological relevance such as aromatic amino acids, nucleic acids, photosynthetic pigments, and steroids (Martínez-Pérez et al., 2017). As a result, a254 -although not representing all DOC present- can be used as reliable proxy for the evolution of DOM concentration during incubation experiments." (P7, L203-208).

Incorporation of responses into the main text: While the authors indicated in the Response document that they added corresponding text to the revised manuscript, I noticed that some responses (e.g., comments 2 and 3 from Reviewer 2) were not incorporated into the main text. Please thoroughly cross-check the revised manuscript to ensure all responses are accurately reflected.

Authors reply: We apologize for this omission. While we initially addressed the reviewer's comment, it was inadvertently removed during the final editing stage, where we significantly reduced the word count. We have now properly reinstated all comments from the first review round.

**Specific Suggestions**

Lines 46-48: The connection between the two sentences is not sufficiently clear.

Authors reply: We agree with this remark, we rewrote the sentence as follows:
"This effect likely arises from reducing the number of active, DOM-consuming bacteria and by providing egestion DOM compounds. At the onset of the productive season, high bacterial growth rates stimulate protistan grazing, which serve as a link between

bacterial biomass and higher trophic levels. However, as grazing pressure increases, protists can also contribute to the accumulation of a fraction of DOM." (P2, L44-48).

Lines 78-89: The section is overly redundant. Consider presenting the purpose first, followed by the main findings in a concise manner.

Author reply: Thank you for this suggestion, we rewrote the paragraph to make it more concise as follows:

"Our study aimed to evaluate the role of protists in transferring bacterial and phytoplankton biomass under different productivity conditions and to assess potential changes in the chromophoric fraction of DOM and FDOM in these scenarios. We found that protistan grazing selectively targeted bacteria under both productivity regimes, and that intense grazing pressure on bacteria may contribute to the short-term accumulation of DOM substances, irrespective of productivity levels. These findings enhance our understanding of the ecological mechanisms influencing carbon flow in marine ecosystems." (P3, L78-84).

Line 101: Revise to "an SBE9plus CTD profiler."

Authors reply: Done (P4, L96).

Lines 104-105: Revise the ion abbreviations to "$NO_3^-$, $NO_2^-$, $NH_4^+$, $PO_4^{3-}$, $SiO_4^{2-}$" for consistency and clarity.

Authors reply: Thank you for your remark. We replaced Si to $SiO_4^{2-}$ in the new version of the manuscript.

Line 107: Specify the pore size of the Whatman GF/F filter and review the manuscript to ensure all filter pore sizes are stated consistently.

Authors reply: Done.

Line 125: Provide the exact pore size of the Whatman polycarbonate filter.

Authors reply: We used a Whatman polycarbonate filter with a pore size of 0.2 µm, as stated in the manuscript (P4, L122).

Line 129: Check the unit, as "$g\ m^2$" seems incorrect or incomplete.

Authors reply: The unit represents grams of fabric per square meter, a standard term for describing a fabric's translucency or light permeability. To enhance clarity, we have revised its definition in the manuscript as "g of fabric per $m^2$." (P5, L126-127).

Line 130: Delete "(N, P, and Si)".

Authors reply: Done.

Lines 198-199: The explanation of HIX provided here is inaccurate. HIX is primarily used to assess the degree of humification in DOM, which reflects its aromaticity and the extent of diagenetic alteration.

Authors reply: We have changed the definition according to the reviewer's suggestion (P6, L195-196).

Line 366: In Figure 8b, the value of R2 is 0.759...

Authors reply: Thank you for catching this error. The correct $R^2$ value is 0.837, and we have now corrected it in both the main text and the figure.

Line 429: Why is the growth rate of heterotrophic bacteria high at all stations?

Authors reply: While our experimental design did not allow us to detect DOC excess or limitation as DOC was only measured at the surface layer and not at the depth where experiments were done (i.e., chlorophyll-a maximum), we can speculate that heterotrophic bacteria were under replete DOC conditions, in addition of being supplemented with inorganic nutrients during experiments. The concentration of DOC at the surface ranged between 52.3 and 95.7 μM, which is among the moderate to higher DOC range in open waters (Hansell et al. 2009, Oceanography). We added some text to clarify this point.

"The moderate to high DOC concentration compared to other open waters (Hansell et al. 2009), appears to support sustained bacteria growth." (P18, L496-497).

Line 467: Clearly state whether ciliates preferentially graze on smaller prey, such as bacteria?

Authors reply: Ciliates do not preferentially prey on bacteria; rather, they can thrive when the available prey are of uniform size. We revised the text to clarify this point.

"Ciliates primarily select prey based on size as their oral diameter limits ingestion. They can thrive on a diet composed exclusively of prey of uniform size, such as bacteria. In contrast, dinoflagellates employ diverse feeding strategies, allowing them access to a broader range of prey sizes." (P17, L478-480).

Lines 491-508: These statements should be revised to improve logical flow and readability, ensuring that the connections between ideas are clearer and the overall message is more coherent.

Authors reply: We agree with this remark. We improved the readability of the paragraph as follows:

"The complexity of DOM, reflected by the prevalence of humic-like compounds ($FDOM_C$, $FDOM_A$, and $FDOM_M$), emerged as a key factor influencing the bacterial utilization of DOM. A net consumption of the DOM fraction, as estimated by a254, was observed when the initial DOM pool contained a significant proportion of low-reactivity compounds, influenced by the phenological status of the sampling site, as indicated by satellite chlorophyll-a trends. However, while stations 21 and 23 had high proportions of refractory compounds, they differed in labile compound content, leading to distinct experimental outcomes. Both stations had low pre-sampling chlorophyll-a, indicating limited recent autochthonous DOM production, yet showed contrasting bacterial activity. In station 23, high bacterial activity increased DOM complexity and net consumption under both treatments, suggesting dependence on an initial labile DOM pool (FDOMT). In contrast, station 21 had low bacterial activity, with the a254 DOM fraction shifting from consumption to accumulation without protists, indicating bacterial reliance on phytoplankton DOM exudates when the DOM pool was predominantly refractory." (P18, L508-519).

Lines 504-505: I didn't quite understand this statement. A low phytoplankton-to-bacteria biomass ratio indicates relatively high bacterial biomass. So, why would bacterial DOM consumption be masked by phytoplankton?

Authors reply: Thank you for identifying this error. The term was misspelled, and the correct ratio used in the manuscript is "bacteria-to-phytoplankton". We have corrected this in the text accordingly. (P18, L517).

Lines 538-541: Awkward sentence, please rephrase

Authors reply: We agree with this remark. We improved the readability of the paragraph as follows:

"Our experimental results indicate that the diversion of bacterial carbon from the DOM cycle by protistan grazing is only partial, as protistan grazing introduces additional DOM, while the reduction in bacterial abundance leaves a fraction of phytoplankton-derived DOM unutilized over short timescales." (P20, L563-566).

Figure 8b: I don't fully understand the positive relationship between the net a254 change and HB:Phyto. A higher HB:Phyto ratio indicates more HB biomass, which should correspond to a greater consumption of DOM. However, the figure shows the opposite trend.

Authors reply: This trend was observed only in the treatment without protists. We interpret this as a high dependence of bacteria on phytoplankton-derived DOM when bacterial biomass is low relative to phytoplankton (i.e., a low HB:Phyto ratio). Under these conditions, bacteria appear to rely more on phytoplankton-derived DOM than on other sources. If this interpretation is correct, the removal of protists—along with the absence of newly produced DOM—would lead to net a254 DOM fraction consumption in the incubation bottle.
We added some text in the discussion section to clarify this issue:

"Low bacteria-to-phytoplankton biomass ratios indicated that bacterial DOM consumption was masked by phytoplankton biomass but became evident without protists. That is, in stations were bacterial biomass is low relative to phytoplankton, bacteria appear to rely more on phytoplankton-derived DOM than on other sources. This effect become apparent upon removal of protists that implies the absence of newly produced DOM (Fig. 8b), in which a net a254 DOM fraction consumption was observed." (P18, L522-526).

**RC2**

I think the manuscript by Lopez-Abbate has significantly improved since the first round of revision. I have some minor comments that I think prevent this manuscript from being accepted in it's current state that are discussed in the public revision section. Mainly, I do not agree with putting heterotrophic bacteria together with phytoplankton in reference to the visual display of the trophic pyramid. Heterotrophic bacteria are secondary consumers and should not be lumped in with phytoplankton as the bottom of a trophic pyramid. I skimmed through the McCauley et al 2018 mansucript they reference and here there are examples (e.g. Cho & Azam 1990, or even more pertinent Gasol et. al. 1997) where heterotrophic bacteria are clearly separated from phytoplankton in a trophic structure. Beyond this, the manuscript has some minor grammar in need of editing. Authors specifically reference Kang 2023 on food webs where phytoplankton and heterotrophic bacteria biomass are summed, specifically because they are eaten by protozooplankton - this manuscript summed them, and did not show food webs where phytoplankton and phytoplankton+heterotrophic bacteria both existed.

López-Abbate et al investigated how phytoplankton and bacterial mortality influence the pool of dissolved organic matter (DOM) on the Patagonian Shelf; a region of the ocean known as a hotspot of carbon sequestration. Dilution experiments revealed that phytoplankton mortality via grazing was greater than their growth rate, while bacterial growth and mortality were relatively balanced. By measuring colored dissolved organic matter (CDOM) and fluorescent dissolved organic matter (fDOM), authors observed a decrease in 'fresh' DOM and an increase in aromatic DOM. How carbon flows through multiple trophic levels is of importance for understanding marine ecosystems, especially in the context of carbon dioxide sequestration. I thank the authors for

returning a heavily revised manuscript that demonstrates a marked improvement from the previous version, however I still have some comments that need to be addressed.

Authors responded to reviewer comments by stating they had edited the text to define what they consider to be a phytoplankton bloom. This seems to be missing from the current version of the manuscript and I believe this should be added to clearly define what they mean by a phytoplankton bloom.

Authors reply: We apologize for this omission. While we initially addressed the reviewer's comment, it was inadvertently removed during the final editing stage, where we significantly reduced the word count. We have now reinstated the definition of a phytoplankton bloom and carefully reviewed the manuscript to ensure that all comments from the first review round are properly incorporated.

I thank the authors for clarifying how they determined their trophic pyramids in figure 4A. I have now read through Kang et al 2023 and McCauley et al 2018. The Kang paper sums phytoplankton and bacteria in ecosystems where they are both prey items for protozooplankton. In these ecosystems, the trophic structure phytoplankton+heterotrophic bacteria, protozooplankton, and mesozooplankton - they do not create a pyramid that has phytoplankton, and phytoplankton+heterotrophic bacteria. The McCauley et al 2018 paper, on the other hand, treats heterotrophic bacteria as a higher trophic level than phytoplankton (see examples cited in text such as Gasol et al 1997 and Cho & Azam 1990). If the authors are citing Kang et al 2023 (which is not in the figure legend, at odds with author's response), they need to remove the phytoplankton from their trophic pyramid and only display Phytoplankton+Heterotrophic bacteria. Otherwise, heterotrophic bacteria are secondary consumers and at a higher trophic level than phytoplankton.

Authors reply: We modified the figure 4a according to the reviewer's suggestion and added the reference to Kang et al. 2023 into the figure's legend.

Minor comments:
Line 35 – 36: unclear phrasing, which microbes had a faster growth rate than the others?

Authors reply: Thank you for your remark, we rewrote the sentence as follows:

"Although phytoplankton biomass was higher than that of bacteria, protists selectively preyed on the faster-growing bacterial population, denoting trophic specificity of grazers." (P2, L35-36).

Lines 38 – 40: the terms top-heavy or bottom-heavy pyramid structure are not clear terms to me in this context, and I am not sure your data measures trophic efficiency, as respiration was not measured in this dataset.

Authors reply: We agree with this remark and have replaced 'trophic efficiency' with 'trophic coupling.' Additionally, we now refer to a top-heavy pyramid as an 'inverted trophic pyramid structure.' However, we have retained the term 'bottom-heavy pyramid' to clearly distinguish it from the less common inverted pyramid. The text now reads as follows:

"High trophic coupling was suggested by the biomass distribution of protistan consumers and their prey, which predominantly exhibited an inverted trophic pyramid structure. An exception to this pattern was observed at the highly productive shelf break front, where a traditional bottom-heavy pyramid emerged, indicating that most phytoplankton evaded protist predation despite evidence of herbivory." (P2, L36-40).

Line 450 – 451: This sentence is speculative, and the citation (Taylor & Landry 2018) is not a study that demonstrates or even discusses this relationship in heterotrophic bacteria. I suggest this be removed.

Authors reply: We appreciate the reviewer's feedback and acknowledge the concern regarding the speculative nature of the sentence. However, we would like to clarify that the phrase 'the suggested mechanism underlying experimental observations' was already present in the original manuscript and remains unchanged in the revised version. Our reasoning aligns with Michel Landry's Enhanced Microbial Loop Hypothesis and its extensions, which we believe provide a solid conceptual framework for our interpretation. Therefore, we prefer to retain the text as it stands unless the reviewer strongly suggests otherwise. Additionally, we agree that the reference to Taylor & Landry (2018) may not be the most appropriate and have accordingly replaced it with Landry et al. (2023) (P17, L452).